# Biofilm microenvironment triggered self-enhancing photodynamic immunomodulatory microneedle for diabetic wound therapy

Li Yang[1], Dan Zhang[1], Wenjing Li[1], Hongbing Lin[1], Chendi Ding[2,3], Qingyun Liu[1], Liangliang Wang[4], Zimu Li[1], Lin Mei [1,2], Hongzhong Chen [1,3] ✉, Yanli Zhao [3] ✉ & Xiaowei Zeng[1] ✉

The treatment of diabetic wounds faces enormous challenges due to complex wound environments, such as infected biofilms, excessive inflammation, and impaired angiogenesis. The critical role of the microenvironment in the chronic diabetic wounds has not been addressed for therapeutic development. Herein, we develop a microneedle (MN) bandage functionalized with dopamine-coated hybrid nanoparticles containing selenium and chlorin e6 (SeC@PA), which is capable of the dual-directional regulation of reactive species (RS) generation, including reactive oxygen species (ROS) and reactive nitrogen species (RNS), in response to the wound microenvironment. The SeC@PA MN bandage can disrupt barriers in wound coverings for efficient SeC@PA delivery. SeC@PA not only depletes endogenous glutathione (GSH) to enhance the anti-biofilm effect of RS, but also degrades GSH in biofilms through cascade reactions to generate more lethal RS for biofilm eradication. SeC@PA acts as an RS scavenger in wound beds with low GSH levels, exerting an anti-inflammatory effect. SeC@PA also promotes the M2-phenotype polarization of macrophages, accelerating wound healing. This self-enhanced, catabolic and dynamic therapy, activated by the wound microenvironment, provides an approach for treating chronic wounds.

Chronic wounds are a common complication of diabetes that affect ~25% of patients with diabetes[1,2]. The annual costs for treating chronic wounds and scars exceed $5 billion, imposing a huge financial burden on patients and the healthcare system[3]. The healing process of diabetic wounds is hindered by biofilms infections, persistent inflammation, and impaired angiogenesis[4–6]. Currently, several therapeutic strategies are available for managing diabetic wounds, including aggressive debridement, surgical angiogenesis, antibacterial therapy, and bioengineered alternative tissue products[7,8]. However, most of these strategies are restricted by the complex pathogenesis of chronic wounds and are not completely effective for the treatment of diabetic wounds. Therefore, there is a need for the development of versatile therapeutic

[1]School of Pharmaceutical Sciences (Shenzhen), Sun Yat-sen University, Shenzhen 518107, China. [2]Tianjin Key Laboratory of Biomedical Materials, Key Laboratory of Biomaterials and Nanotechnology for Cancer Immunotherapy, Institute of Biomedical Engineering, Chinese Academy of Medical Sciences & Peking Union Medical College, Tianjin 300192, China. [3]School of Chemistry, Chemical Engineering and Biotechnology, Nanyang Technological University, 21 Nanyang Link, Singapore 637371, Singapore. [4]School of Public Health (Shenzhen), Sun Yat-sen University, Shenzhen 518107, China. ✉e-mail: chenhzh58@mail.sysu.edu.cn; zhaoyanli@ntu.edu.sg; zengxw23@mail.sysu.edu.cn

strategies encompassing antibacterial, anti-inflammatory, and pro-angiogenesis effects to address the treatment of diabetic wounds.

Chronic wounds, including diabetic wounds, are frequently infected with biofilms[9,10]. Bacteria are protected by extracellular polymeric substances (EPS) of the biofilms that form a strong barrier against antibiotics and immune cells[11,12]. Because of being packed in EPS, these biofilms exhibit specific microenvironments in wound beds, such as low pH and high levels of glutathione (GSH)[13,14]. Therefore, exploiting the special characters of the biofilms microenvironment is a promising strategy to prevent biofilms infections. Photodynamic therapy (PDT), a minimally invasive procedure that does not involve issues of drug resistance, has received significant attention as an antibacterial approach[15–17]. There are challenges associated with the use of PDT to effectively treat biofilms infections, as the ROS generated by photosensitizers (PS) can be depleted by the high levels of GSH in biofilms[18]. To address these issues, most studies focus on the depletion of endogenous GSH by mediators such as nitric oxide (NO)[12] or $Cu^{2+}$.[19] However, the strategy of exploiting high GSH level in biofilms to enhance PDT has been barely reported. Previous studies have shown that selenite ($SeO_3^{2-}$) could target high GSH levels in tumor cells to produce ROS, including $O_2^{\bullet-}$ and $\bullet OH$[20–22]. In contrast, ROS production, which is lethal to cells, is not triggered in normal cells with low GSH levels, thereby protecting cells from apoptosis[22]. Based on this inspiration, we propose a strategy to effectively amplify PDT by depleting endogenous GSH, and generate active molecules that are

more lethal to biofilms by utilizing endogenous GSH, thus facilitating the eradication of biofilms.

Another point of concern is that ROS is a double-edged sword[23]. Wound beds covered by biofilms are excessively inflammatory, and oxidative stress in wounds can be aggravated by the ROS produced during PDT to destroy biofilms, leading to impaired wound healing[24]. Therefore, the prompt elimination of ROS during PDT is crucial to prevent further exacerbation of wound inflammation, as well as enhance the antioxidative capacity of the wound beds for providing an anti-inflammatory effect from the outside to the inside. Moreover, an important factor affecting the efficacy of therapeutics is their point of delivery. The importance of the role of the delivery point in promoting wound healing has been well studied[9]. Chronic wounds are typically covered with inflammatory exudates and biofilms[25], which constitute a physicochemical barrier hindering the penetration of topical therapeutic agents into the underlying live tissues. Thus, when drugs are applied topically, their local bioavailability is reduced, as expected. A delivery system that can disrupt the physiochemical barriers created by inflammatory exudates and biofilms, shorten the travel distance of therapeutics by directly delivering them to wound tissue can improve drug efficacy.

In this work, we develop a multifunctional microneedle (MN) bandage for diabetic wound treatment. As shown in Fig. 1a, dopamine-coated hybrid nanoparticles (SeC@PA) containing selenium (Se) and chlorin e6 (Ce6) are firstly prepared, and then modified with L-arginine

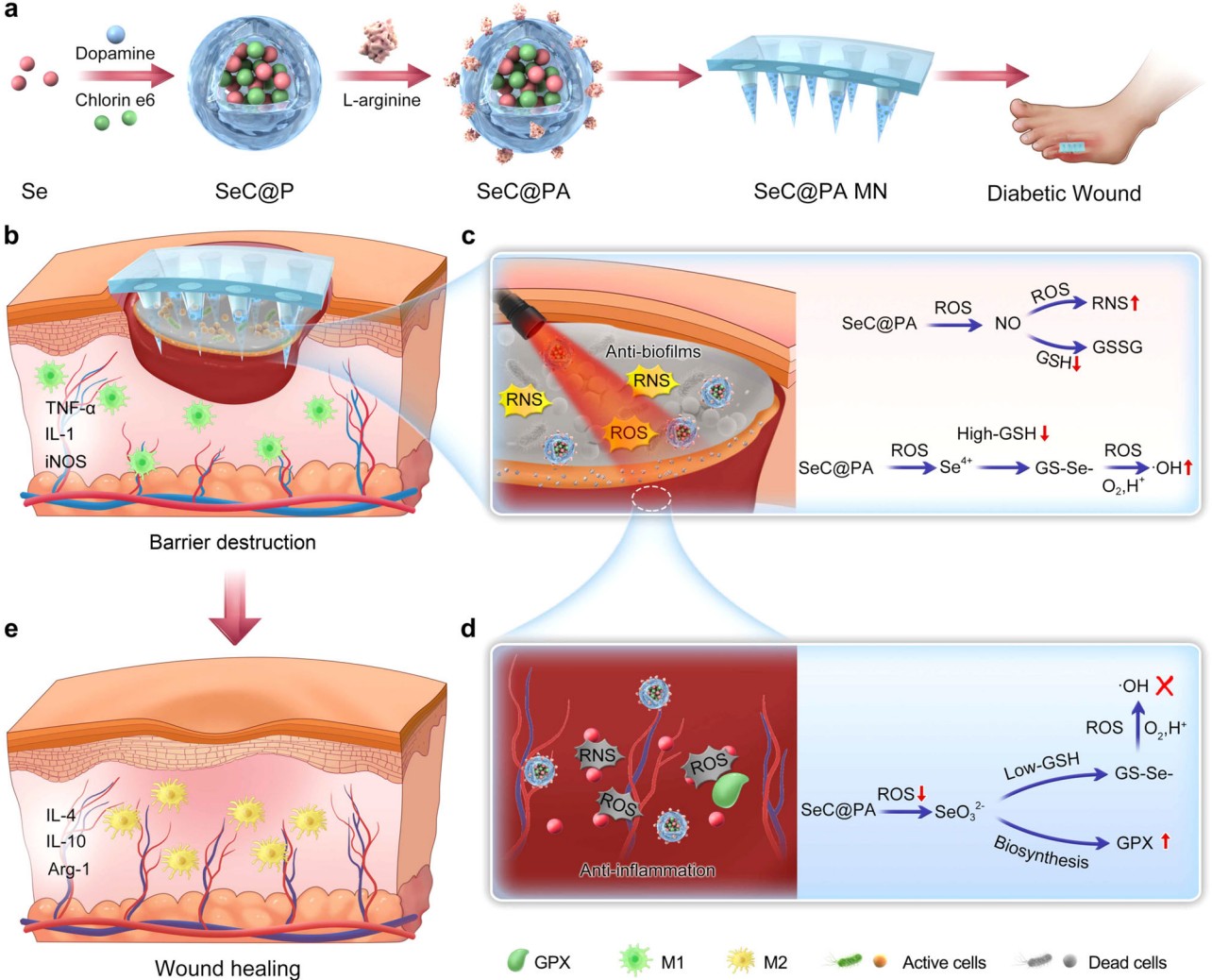

**Fig. 1 | Self-enhancing catabolic dynamic strategy for diabetic wound therapy. a** Scheme of fabrication process of SeC@PA MN bandage. **b–e** Scheme of barrier destruction and promoting wound healing mechanisms of SeC@PA MN.

(LA) on the surface. The MN bandage is loaded with SeC@PA to obtain SeC@PA MN. When the SeC@PA MN is applied to diabetic wounds, the physiochemical barriers are broken, and SeC@PA are effectively delivered to the biofilms and wound live tissue (Fig. 1b). We hypothesize that after delivery into the biofilm-infected sites, SeC@PA can not only deplete GSH and generate reactive nitrogen species (RNS) to enhance the anti-biofilms effect of PDT, but also utilize the high levels of GSH in biofilms to generate hydroxyl radical (•OH) through a series of catalytic cascade reactions (Fig. 1c). Conversely, after delivery into the highly inflammatory wound tissue, SeC@PA exhibits potent antioxidant effects by rapidly scavenging RS and enhancing glutathione peroxidase (GPX) activity (Fig. 1d). Moreover, SeC@PA can effectively promote macrophage polarization toward the M2-phenotype, thereby facilitating wound healing (Fig. 1e). The bidirectional regulation of RS by SeC@PA through different GSH levels is demonstrated both in vitro and in vivo. The therapeutic efficacy of SeC@PA MN is studied in a mouse model of diabetic wounds. Our study highlights a self-augmented, catabolic, and dynamic therapy to effectively manage chronic diabetic wounds.

## Results

### Preparation and characterization of SeC@PA and SeC@PA MN bandage

In this study, Se was first prepared following an established method[26,27]. Transmission electron microscopy (TEM) images indicated that Se particles were uniform with an average diameter of $38.42 \pm 0.85$ nm (Supplementary Fig. 1a, d). Dopamine (DA) was used to coat the surface of Se and encapsulate the photosensitizer (Ce6) to obtain SeC@P. LA was adsorbed on the surface of SeC@P by electrostatic interaction to form SeC@PA. The characterization results are shown in Fig. 2. Product morphologies were observed using TEM. SeC@P exhibited a spheroid morphology (Supplementary Fig. 1e) with good dispersity similar to that of Se@PA (Fig. 2a). The hydrodynamic diameters of SeC@P and SeC@PA were $90.00 \pm 1.82$ nm and $108.47 \pm 1.01$ nm (Supplementary Fig. 1b, c), respectively. The zeta potentials indicated that SeC@P had a negative charge and SeC@PA had a positive charge (Supplementary Fig. 1f), suggesting that the positively charged LA was successfully introduced on the surface. The successful assembly of SeC@PA was also validated using the scanning electron microscopy (SEM) energy dispersive spectrometer (EDS) (Supplementary Fig. 2) and X-ray photoelectron spectroscopy (XPS) (Supplementary Fig. 3). The results revealed that SeC@PA was composed of not only the basic elements C, O, and N, but also Se. The loading efficiency of Se, Ce6, and LA in SeC@PA was calculated as $23.23 \pm 4.11\%$, $8.40 \pm 4.38\%$, and $12.60 \pm 3.79\%$, respectively (Supplementary Table 1).

It was expected that the delivery of SeC@PA to biofilms would result in the rapid degradation of the PDA layer, leading to the release of Ce6 and Se. This release would trigger a cascade reaction, generating an RS storm to eradicate the biofilms. Therefore, we studied the release behavior of Ce6 and Se from SeC@PA under different pH conditions. The release of Ce6 from SeC@PA was accelerated in a weakly acidic environment (pH = 5.5) (Supplementary Fig. 4) and Se was also successfully released in pH = 5.5 condition (Supplementary Fig. 5). Since the biofilm microenvironment is weakly acidic[28,29], Se and Ce6 were effectively released to the action point when SeC@PA was delivered to the biofilms. Next, we investigated whether singlet oxygen ($^1O_2$) produced by Ce6 could oxidize LA adsorbed on the surface to produce NO for GSH depletion and RNS formation, thereby amplifying the RS anti-biofilm effect. The studies showed that NO was successfully detected under irradiation (Fig. 2b, c), which positively correlated with irradiation intensity and SeC@PA concentration. We also examined that $^1O_2$ produced by Ce6 during irradiation could oxidize the Se released from SeC@PA to $Se^{4+}$ in the first step of the cascade. High-resolution XPS revealed that $Se3d$ from SeC@PA with irradiation

(SeC@PA(+)) consisted of $Se^0$ (54.73 eV and 55.53 eV) and $Se^{4+}$ (58.20 eV and 59.06 eV) (Fig. 2f). In contrast, Se and SeC@PA without irradiation only contained $Se^0$ (54.73 eV and 55.53 eV) (Supplementary Fig. 6), indicating that the Se released from SeC@PA could be oxidized to $Se^{4+}$ under irradiation.

To verify the ability of SeC@PA to bidirectionally regulate RS at different GSH levels in vitro, the fluorescent probe 2,7-dichloro-fluorescein diacetate (DCF-DA) was used to evaluate RS generation. As shown in Fig. 2d, abundant RS was produced by C@PA under irradiation (C@PA(+)) in PBS solution (pH = 5.5) without GSH. At the same time, RS generated by SeC@PA(+) was significantly reduced, proving that Se could eliminate RS in the absence of GSH. Nevertheless, when C@PA(+) was co-incubated with 8 mM GSH (high GSH levels in biofilms), RS levels were dramatically decreased due to the depletion by GSH. Surprisingly, unlike C@PA(+), SeC@PA(+) exhibited the highest RS levels in 8 mM GSH solution, which was attributed to Se producing RS with more oxidative activity by decomposing GSH. In vitro experiment results demonstrated that SeC@PA(+) could more quickly deplete GSH during irradiation compared with C@PA(+) (Fig. 2i). Subsequently, we explored the impact of varying concentrations of GSH on the RS generated by SeC@PA (+) and C@PA (+). As exhibited in Fig. 2e, the intensity of RS in the C@PA(+) group was observed to decrease with increasing GSH concentration, which was attributed to the scavenging effect of GSH on the RS. In contrast, SeC@PA(+) played a role in scavenging RS at low GSH concentrations (≤0.4 mM), while enhancing RS intensity at high concentrations (≥ 0.8 mM). A turning point was clearly observed, which indicated that SeC@PA successfully achieved bidirectional regulation of RS. When a low-level GSH (2 μm), for simulating the GSH concentration in healthy tissue, was introduced into the system, similar results were obtained as compared with that without GSH, where SeC@PA(+) mainly acted as a RS scavenger (Supplementary Fig. 7). Electron paramagnetic resonance (EPR) (Fig. 2g, h) and methylene blue (MB) (Supplementary Fig. 8) degradation were used to determine •OH production. The results showed that SeC@PA(+) could produce active •OH only when GSH levels were high. These results suggested that at high GSH levels, SeC@PA(+) participated in GSH degradation and led to higher RS levels, whereas RS-scavenging ability was exhibited at low GSH levels, thereby achieving bidirectional RS regulation.

We successfully prepared the SeC@PA-loaded MN bandage (Fig. 2j). The SeC@PA MN bandage exhibited rapid dissolution performance in PBS (Supplementary Fig. 9), indicating that the application time of the SeC@PA MN bandage was short. To assess the delivery efficiency of SeC@PA, we assembled an in vitro model mimicking the wound barrier. The in vitro model consisted of biofilms formed by *Escherichia coli* (EC) expressing the green fluorescent protein (GFP) in an agarose gel. A rhodamine-loaded SeR@PA MN bandage was applied on the top of the mold to simulate topical drug delivery. At the same time, SeR@PA nanoparticles (NPs) were added to the top of the mold for comparison. The results (Fig. 2k) demonstrated the remarkable capability of the MN bandage to penetrate barriers and directly deliver SeR@PA to the desired target region. In contrast, SeR@PA NPs alone exhibited limited penetration, primarily confined to the surface of the barrier, thereby impeding its ability to reach the deeper target regions. Collectively, these results suggested that the MN bandages could overcome the obstacles of local drug delivery in chronic wounds, delivering the drug directly to the site of action for therapeutic effect.

### Anti-biofilm property of SeC@PA

For anti-biofilm evaluation, *Staphylococcus aureus* (SA) and *Pseudomonas aeruginosa* (PA) were used as the representative strains as they are the most common species isolated from chronic wound biofilms[30]. The antibacterial activity and cell viability were first investigated with different concentrations of SeC@PA(+). As shown in Supplementary Fig. 10, the antibacterial activity was enhanced with an increased

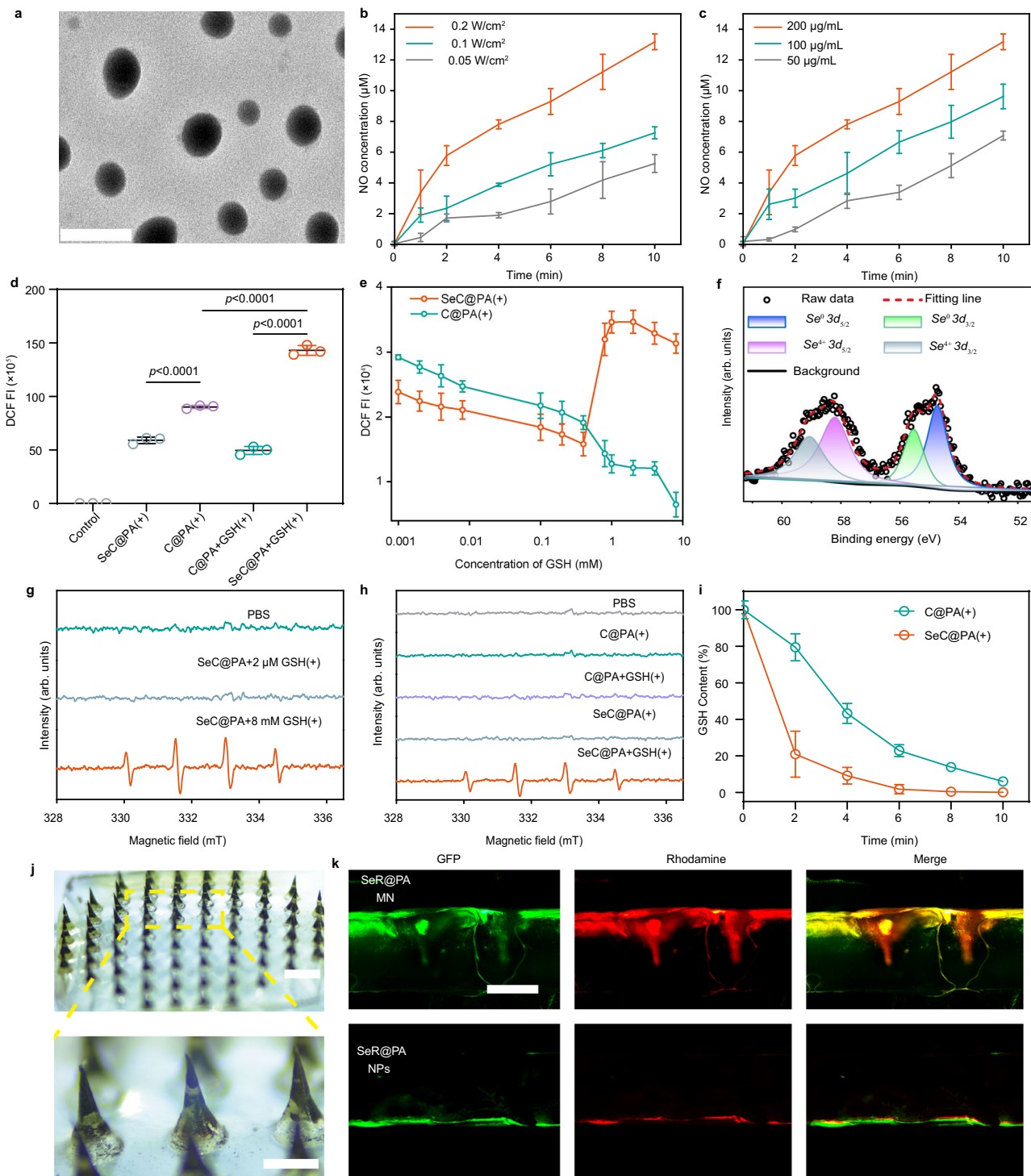

**Fig. 2 | Characterizations of SeC@PA and SeC@PA MN. a** TEM images of SeC@PA. Scale bar is 200 nm. Two independent experiments were performed and representative results are shown. **b** Concentration of NO produced by SeC@PA (200 µg/mL) with different irradiation intensities at 660 nm ($n$ = 3 independent samples; mean ± SD). **c** Concentration of NO produced by SeC@PA with different concentrations under 660 nm irradiation (0.2 W/cm²) ($n$ = 3 independent samples; mean ± SD). **d** DCF fluorescence intensity of C@PA(+) and SeC@PA(+) in the presence or absence of GSH ($n$ = 3 independent samples; mean ± SD). **e** Effect of different GSH concentrations on the RS produced by SeC@PA(+) and C@PA(+) ($n$ = 4 independent samples; mean ± SD). **f** High-resolution *Se 3d* XPS spectra of SeC@PA(+). **g** EPR analysis of •OH production in each group with different GSH levels. DMPO was used as the spin-trapping agent. **h** EPR analysis of •OH production in each group with different treatment. DMPO was used as the spin-trapping agent.

**i** In vitro degradation curves of GSH in C@PA(+) and SeC@PA(+) ($n$ = 3 independent samples; mean ± SD). **j** Appearance of SeC@PA MN. Scale bar is 500 µm. Two independent experiments were performed and representative results are shown. **k** Penetration depth of SeR@PA NPs and SeR@PA MN to biofilms and agarose gels in vitro. Scale bar is 500 µm. Two independent experiments were performed and representative results are shown. Statistical significance was analyzed via one-way ANOVA with a Tukey post-hoc test. Source data are provided as a Source Data file. C@PA: Ce6-PDA-LA nanoparticles; SeC@PA: Se-Ce6-PDA-LA nanoparticles; C@PA(+): Ce6-PDA-LA nanoparticles under 660 nm irradiation (0.2 W/cm²) for 3 min; SeC@PA(+): Se-Ce6-PDA-LA nanoparticles under 660 nm irradiation (0.2 W/cm²) for 3 min; SeC@PA MN: microneedle containing Se-Ce6-PDA-LA nanoparticles; SeR@PA NPs: solution containing Se-rhodamine-PDA-LA nanoparticles; SeR@PA MN: microneedle containing Se-rhodamine-PDA-LA nanoparticles.

concentration of SeC@PA(+) and also accompanied by stronger cytotoxicity. To balance the antibacterial activity and cytotoxicity, we selected a concentration of SeC@PA containing 20 µg/mL of Se, 7.2 µg/mL of Ce6, and 10.8 µg/mL of LA for follow-up studies. Next, we explored the bactericidal effects of different agents on SA and PA. Both C@PA(+) and SeC@PA(+) had strong antibacterial effects (Supplementary Fig. 11), demonstrating the antimicrobial potential by targeting RS.

To elucidate the "RS storm" strategy, we investigated the anti-biofilm contribution of each kind of RS ($^1O_2$, RNS, and •OH) produced in SeC@PA(+) by RS superposition method, which was specifically described in the method section. $^1O_2$ was generated in C@P + 8 mM GSH group, C@PA + 8 mM GSH group, and SeC@PA + 8 mM GSH group, RNS was produced in C@PA + 8 mM GSH group, and SeC@PA + 8 mM GSH group, while •OH was only generated in SeC@PA + 8 mM GSH group, and the intensity of each RS had no obvious difference (Supplementary Fig. 12 and Supplementary Fig. 13a). Next, we explored the production of RS by different groups in the biofilms. As presented in Supplementary Fig. 13b, d, DCF fluorescence intensity in SA biofilm was SeC@PA(+) group > C@PA(+) group > C@P(+) group, due to the addition of RS. Simultaneously, the anti-biofilm effects of different groups were investigated. The live/dead biofilm images, crystal violet staining, and bactericidal results are shown in Supplementary Fig. 13c, e, f, respectively. The results showed that the biofilm in the C@P(+)

group remained relatively intact and only exhibited weak anti-biofilm effects. In contrast, the anti-biofilm effect was obviously enhanced in the C@PA(+) group with compromised biofilm integrity, which should be due to the reason that the addition of RNS resulted in a stronger anti-biofilm effect. Among all the groups, SeC@PA(+) group showed the strongest anti-biofilm activity, which benefited from the generation of RNS and •OH. These results were consistent with the RS intensity results in the biofilm. Therefore, three types of RS generated by SeC@PA were involved in the anti-biofilm mechanism in this study, and this strategy is termed as "RS storm". The anti-biofilm contributions of the three RS were quantified through crystal violet staining experiments (represented by the biomass of the biofilm). As shown in Supplementary Fig. 14, the contribution of $^1O_2$, RNS and •OH against biofilm was 18.5%, 37.6%, and 30.2%, respectively.

The anti-biofilm efficacy of different treatments against single-species biofilms formed by SA and PA is shown in Fig. 3 and Supplementary Fig. 15. For typical live/dead biofilm images shown in Fig. 3a, Supplementary Figs. 10a, 16, and 17, the biofilms in the control group and C@PA group were relatively intact, and almost all viable bacteria were stained green. In comparison, the Se@PA and SeC@PA groups exhibited slight anti-biofilm activity due to the antibacterial activity of Se (Supplementary Fig. 18). In the C@PA(+) group, dead bacteria were stained red, and large-scale bacterial death was observed. Remarkably, the SeC@PA(+) group exhibited the lowest levels of viable bacteria,

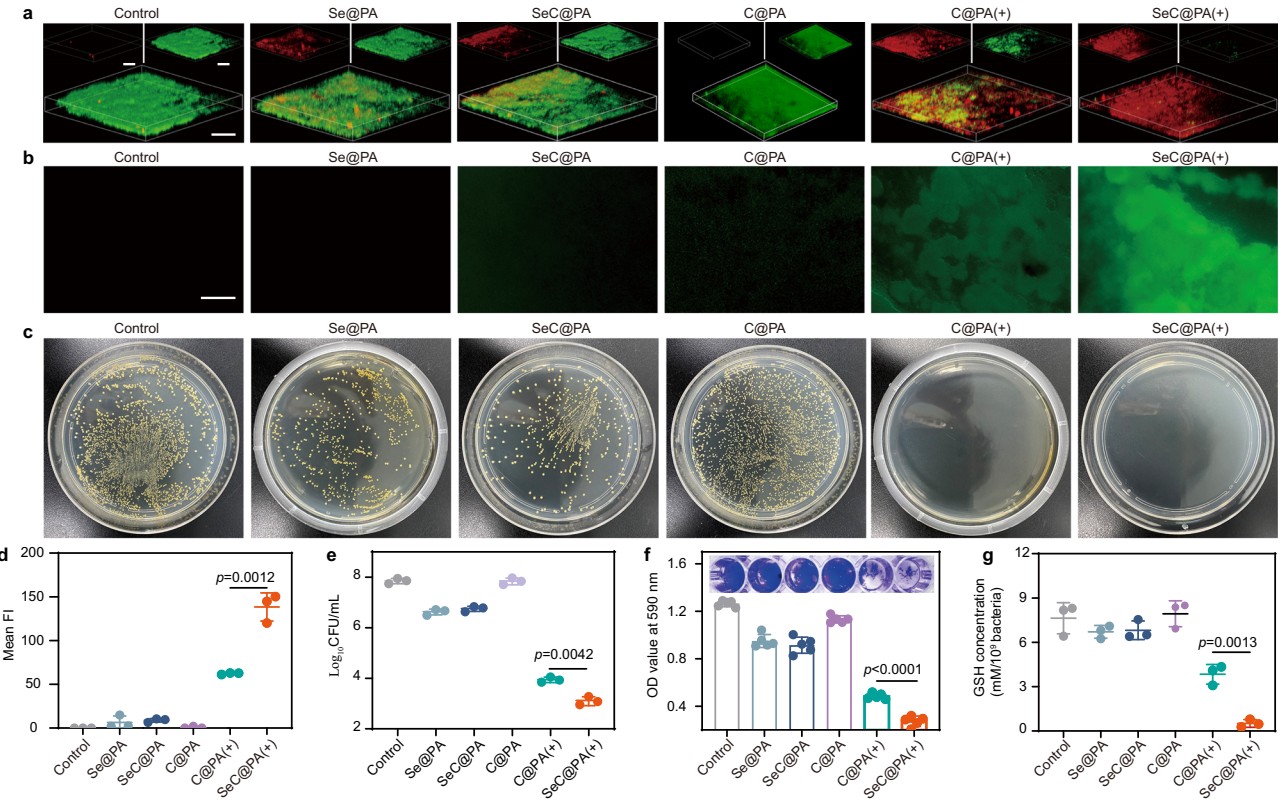

**Fig. 3 | Anti-biofilms effect of nanoparticles. a** Live/dead stain of SA biofilms with different treatments under 660 nm irradiation (0.2 W/cm²) for 3 min. Scale bar is 100 µm. Three independent experiments were performed and representative results are shown. **b** Detection of RS in the SA biofilm incubated with different nanoparticles after 660 nm irradiation (0.2 W/cm²) for 3 min. Scale bar is 500 µm. Three independent experiments were performed and representative results are shown. **c** Representative optical images of colony-forming units for SA biofilm suspension after different treatments. Three independent experiments were performed and representative results are shown. **d** Semiquantitative statistics of the mean fluorescence intensity of DCF in SA biofilms treated with different groups (n = 3 biologically independent samples; mean ± SD). **e** Bactericidal results of

different nanocarriers characterized by the standard plate counting assay (n = 3 biologically independent samples; mean ± SD). **f** Representative images of crystal violet staining and absorbance values at 590 nm after treatment with different nanoparticles (n = 5 biologically independent samples; mean ± SD). **g** GSH concentration in SA biofilms after being treated with different nanoparticles (n = 3 biologically independent samples; mean ± SD). Statistical significance was analyzed via one-way ANOVA with a Tukey post-hoc test. Source data are provided as a Source Data file. C@PA: Ce6-PDA-LA nanoparticles; Se@PA: Se-PDA-LA nanoparticles; SeC@PA: Se-Ce6-PDA-LA nanoparticles; C@PA(+): Ce6-PDA-LA nanoparticles under 660 nm irradiation (0.2 W/cm²) for 3 min; SeC@PA(+): Se-Ce6-PDA-LA nanoparticles under 660 nm irradiation (0.2 W/cm²) for 3 min.

almost entirely covered by dead bacteria. Thus, the SeC@PA(+) group exhibited the best anti-biofilm effect among all groups. Figure 3c and Supplementary Fig. 15c depicted the colony-forming units (CFU) of SA and PA in agar plates, respectively. The quantification of CFUs for two types of single-species biofilm (Fig. 3e and Supplementary Fig. 15g) exhibited similar results for live/dead staining. The CFU values in the SeC@PA(+) group were reduced by 4, 3, and 1 orders of magnitude compared to control group, Se@PA group, and C@PA(+) group, respectively. Similarly, the lowest crystal violet absorbance value was detected in the SeC@PA(+) group for the crystal violet staining experiment (Fig. 3f and Supplementary Fig. 15e), indicating the lowest number of viable bacteria in this group.

To explore whether the SeC@PA(+) group could eradicate biofilms by decomposing GSH to generate higher levels of RS, the GSH concentration and RS in biofilms after different treatments were measured. The green fluorescent probe DCF-DA was used to evaluate RS generation in SA and PA biofilms. As shown in Fig. 3b, d, compared with the C@PA(+) group, the SeC@PA(+) group exhibited higher RS activity in SA biofilms. Whereas, in the group of biofilms incubated with Se@PA or SeC@PA without irradiation, the green fluorescence of RS was weak. These results suggested that SeC@PA(+) could promote RS generation in SA biofilms. Simultaneously, we measured the changes of GSH in the biofilms after treatments (Fig. 3g). As opposed to the groups without laser irradiation, the C@PA(+) and SeC@PA(+) groups could significantly reduce GSH levels in the biofilms by producing NO

after laser irradiation, and GSH levels in SeC@PA(+) group was lower than that in the C@PA(+) group. These results were consistent with the findings obtained in in vitro experiments. Interestingly, the ability of SeC@PA(+) to eliminate GSH was significantly stronger than that of C@PA(+), indicating that Se was involved in the degradation of GSH after irradiation. Similar results were observed for PA biofilms (Supplementary Fig. 15b, d, f). The results indicated that GSH in the biofilm could be decomposed and scavenged by SeC@PA(+) to generate RS storm for biofilms eradication.

## Antioxidant Properties of SeC@PA

RS produced by anti-biofilms further exacerbates the inflammatory response in wound beds. Therefore, the ideal therapeutic effect is that in the case of low levels of GSH, SeC@PA can not only scavenge the RS generated by the anti-biofilm but also eliminate RS generated by wound tissue. Human umbilical vein endothelial cells (HUVECs) were employed to validate the antioxidant properties of SeC@PA as HUVECs are extremely sensitive to RS. As illustrated in Fig. 4a, b, the SeC@PA(+) group demonstrated significantly lower levels of RS compared to the C@PA(+) group. Results from flow cytometry exhibited similar findings (Fig. 4c), indicating that SeC@PA could be an excellent RS scavenger. This behavior of SeC@PA was diametrically opposite to that in biofilms with high GSH levels. Glutathione peroxidase (GPX) is an important peroxidase widely presented in the body, and Se is a component of the GPX enzyme system[31,32]. To evaluate

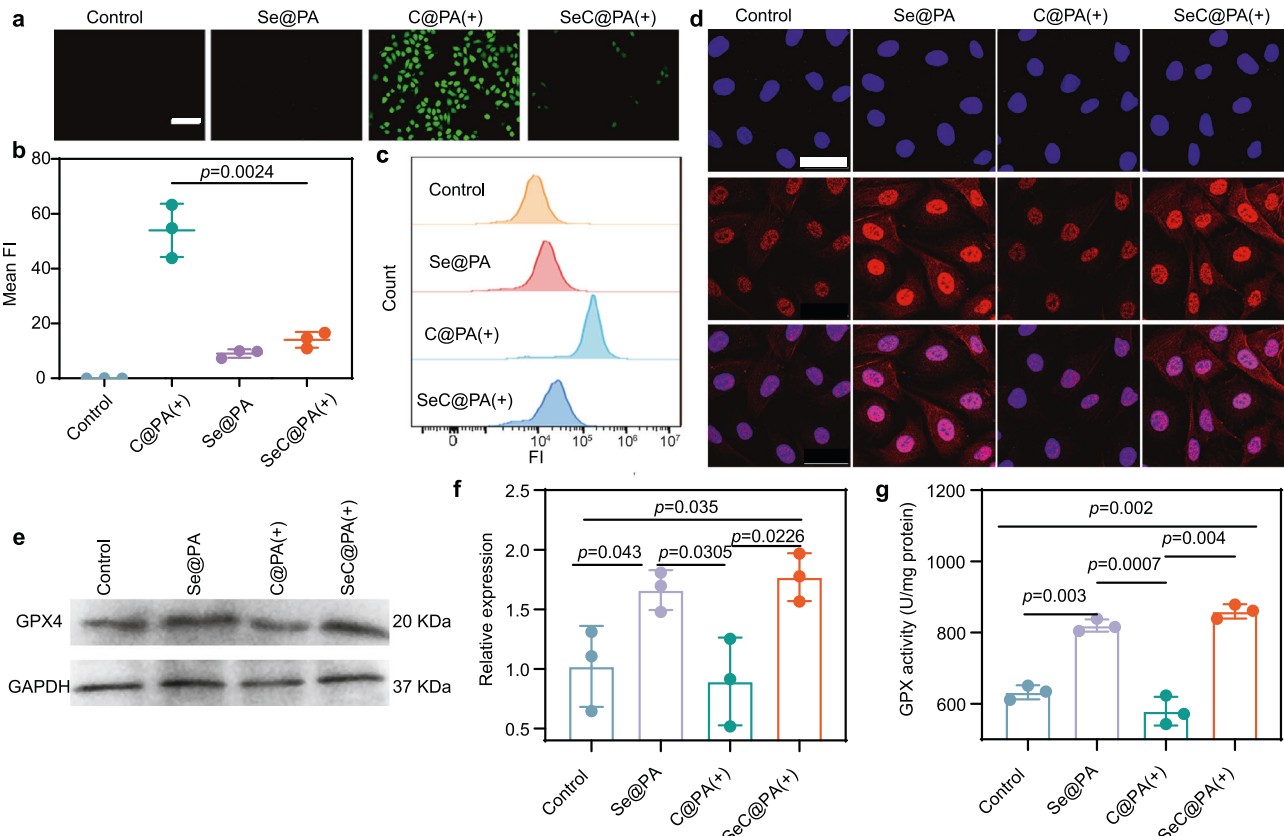

**Fig. 4 | Anti-inflammatory effect of nanoparticles. a, b** Detection of RS by fluorescence of DCF in the HUVECs incubated with different nanoparticles under 660 nm irradiation (0.2 W/cm²) for 3 min (n = 3 biologically independent samples; mean ± SD). Scale bar is 500 μm. **c** Intracellular DCF fluorescence intensity of HUVECs with different groups analyzed by flow cytometry. **d** Representative images of GPX4 staining of HUVECs after different treatments. Scale bar is 200 μm. Three independent experiments were performed and representative results are shown. **e, f** Expression level of GPX4 in the HUVECs determined by western blotting

after treatments (n = 3 biologically independent samples; mean ± SD). The experiment in **e** was repeated three times with similar results. **g** Activity of GPX4 treated with different groups (n = 3 biologically independent samples; mean ± SD). Statistical significance was analyzed via one-way ANOVA with a Tukey post-hoc test. Source data are provided as a Source Data file. Se@PA: Se-PDA-LA nanoparticles; C@PA(+): Ce6-PDA-LA nanoparticles under 660 nm irradiation (0.2 W/cm²) for 3 min; SeC@PA(+): Se-Ce6-PDA-LA nanoparticles under 660 nm irradiation (0.2 W/cm²) for 3 min.

whether SeC@PA could promote the level and activity of GPX through biosynthesis and enhance the antioxidant capacity of wounds, we studied the level and activity of GPX4 after treatment. We labeled GPX4 in HUVECs with red fluorescence and observed them using laser confocal microscopy. As shown in Fig. 4d, the red fluorescence in the cytoplasm and nucleus was significantly enhanced in Se@PA and SeC@PA(+) treated groups compared to the control and C@PA(+) groups, suggesting an increase in GPX4 levels. We further analyzed GPX4 levels using western blotting, and the relative expression of GPX4 in the Se@PA and SeC@PA(+) groups was 1.64- and 1.79-fold higher than that in the control group, respectively (Fig. 4e, f). Subsequently, we examined GPX activity in different treatment groups. GPX activity in the SeC@PA(+) group was 1.74-fold higher than that in the control group, whereas GPX activity in the Se@PA group was 1.30-fold higher than that in the control group (Fig. 4g). Notably, application of C@PA(+) significantly reduced GPX activity compared to the control group, indicating that RS impaired GPX activity, which was reversed by SeC@PA.

## Macrophage polarization–promoting properties of SeC@PA

Macrophages play a central role in regulating inflammation and promoting wound healing and undergo polarization into the M1- and M2-phenotypes[33–35]. Chronic wounds are infiltrated by M1-phenotype macrophages, resulting in a persistent inflammatory response that delays wound healing[36,37]. It has been proven that M2-phenotype macrophages play a key role in regulating inflammation, promoting angiogenesis, and wound healing[38,39]. Hence, enhancing macrophage polarization to the M2 phenotype provides an opportunity to accelerate chronic wound healing. We first used the characteristic surface biomarkers CCR7 and CD206 to identify M1- and M2-phenotype macrophages and determined the effects of different treatments on macrophage polarization[40,41]. After incubation with Se@PA and SeC@PA(+), a more pronounced green fluorescence of CD206 staining was observed compared with that in the other groups (Supplementary Fig. 19). Subsequently, flow cytometry was employed to determine the ratio of M1- and M2-phenotype macrophages by quantifying CCR7 and CD206 expression (Fig. 5a, Supplementary Figs. 20 and 21). After treatment with Se@PA and SeC@PA(+), the percentage of CD206+ M2-phenotype cells was significantly increased compared with that in the other groups, whereas the percentage of CCR7+ M1-phenotype cells were obviously decreased compared with treatment with C@PA(+).

Additionally, the expression of M2-phenotype macrophage marker genes, including *CD206*, *interleukin (IL)-4*, *IL-10*, and *Arg-1*, improved significantly after treatment with Se@PA and SeC@PA(+). In contrast, the M1-phenotype macrophage marker genes, including *CD86*, *IL-1*, *inducible nitric oxide synthase (iNOS)*, and *tumor necrosis factor (TNF)-α*, were remarkably decreased in the Se@PA and SeC@PA(+) groups (Supplementary Fig. 22). The representative cytokine detection results of M2-like macrophages showed that IL-4 and IL-10 levels increased significantly and that IL-1 and TNF-α levels decreased considerably after SeC@PA(+) treatment (Fig. 5n–q). These results indicated that SeC@PA(+) could effectively activate macrophage polarization to the M2 phenotype.

To further explore the mechanism by which SeC@PA(+) promotes the polarization of macrophage to the M2-phenotype, we used quantitative real-time PCR (QPCR) and western blotting to elucidate the signaling pathway of SeC@PA(+). Previous studies have shown that Se levels can affect IL-4-promoted M2-macrophage polarization[42]. Therefore, we hypothesized that SeC@PA(+) could regulate the expression of certain signals in the IL-4 pathway and promote macrophages toward M2-type polarization. IL-4-mediated M2-macrophage polarization has been well established through a Janus kinase type 1 (JAK1) signal transducer and activator of the transcription 6 (STAT6) pathway[43]. We first investigated the

expression of *STAT6* mRNA after incubation with SeC@PA(+) and found a significant increase in its levels (Supplementary Fig. 23a). Notably, the mannose receptor (MR)-mediated polarization of macrophages toward the M2 type also increases *STAT6* expression through the ERK-STAT6 pathway[6]. Therefore, we used western blotting to further confirm the mechanism of SeC@PA(+) in promoting macrophage polarization. The findings (Fig. 5b–e and Supplementary Fig. 24) showed that SeC@PA(+) significantly upregulated the levels of phosphorylated JAK (p-JAK1) and phosphorylated STAT6 (p-STAT6) but had no significant effect on phosphorylated ERK (p-ERK) levels, indicating that SeC@PA(+) activation occurred via the JAK-STAT6 pathway (Supplementary Fig. 23b).

## Promotion of cell migration and angiogenesis by SeC@PA

The migration and tube formation in HUVECs cultured over SeC@PA(+) was evaluated (Fig. 5f–i and Supplementary Fig. 25). Unsurprisingly, cells in the C@PA(+) group were neither migrated nor angiogenesis was detected because RS production impairs the function of HUVECs. Conversely, after incubation with SeC@PA(+), cell migration and angiogenesis were restored due to the scavenging effect of RS. However, SeC@PA(+) treatment did not result in a significant increase in the migration and angiogenesis of HUVECs compared with those in the control group. Considering that the polarization of macrophages into the M2 type promotes wound healing, we further explored the influence of adding RAW264.7 cells on HUVECs behavior. Surprisingly, after adding RAW264.7 cells, SeC@PA(+) treatment significantly promoted the migration and tube-forming ability of HUVECs compared with those in the other groups. To determine whether this phenomenon was attributed to SeC@PA(+) stimulating RAW264.7 cells to secrete certain factors regulating the behavior of HUVECs, we determined the levels of vascular endothelial growth factor (VEGF) in cells after SeC@PA(+) treatment. As shown in Fig. 5j–m, the levels of VEGF in RAW264.7 cells were significantly increased compared with those in the control group. Conversely, the levels of VEGF in HUVECs treated with SeC@PA (+) group exhibited no obvious difference from those in the control group, indicating that SeC@PA(+) promoted RAW264.7 cells to secrete VEGF, thereby enhancing cell migration and tube-formation abilities of HUVECs. Overall, SeC@PA(+) could promote wound healing by regulating the polarization of macrophages to the M2 type, thereby secreting VEGF, which promotes migration and tube formation in HUVECs.

## Treatment of non-biofilm-infected wounds in diabetic mice using SeC@PA bandage

Diabetic chronic wounds without infected biofilms were employed to verify the direct RS-scavenging effect of SeC@PA MN on wound tissues. The MN groups containing Se (Se@PA MN, Se@PA MN(+), SeC@PA MN, and SeC@PA MN(+) group) exhibited much higher wound healing rates compared to those without Se (Control, Control(+), C@PA MN, and C@PA MN(+) group), as demonstrated in Fig. 6a, f. This result could be attributed to the potent antioxidant capacity of Se. Conversely, the C@PA MN (+) group showed impaired wound healing due to aggravated oxidative stress caused by RS production, as shown in Fig. 6b, g. Interestingly, the level of RS in the SeC@PA MN (+) group was lower than that in the C@PA MN (+) group, suggesting a potential role for SeC@PA MN (+) in RS scavenging in wound tissues. Hematoxylin-eosin (HE) staining (Fig. 6c), Masson's trichrome staining (Fig. 6d, h), and CD31 immunofluorescence (Fig. 6e, i) revealed that the Se-containing MN groups exhibited more regular skin structure, enhanced collagen deposition, and increased blood vessels density. Collectively, in the non-biofilm-infected diabetic chronic wound, Se-containing MN could reduce inflammation by scavenging RS, while promoting wound healing through collagen deposition and angiogenesis.

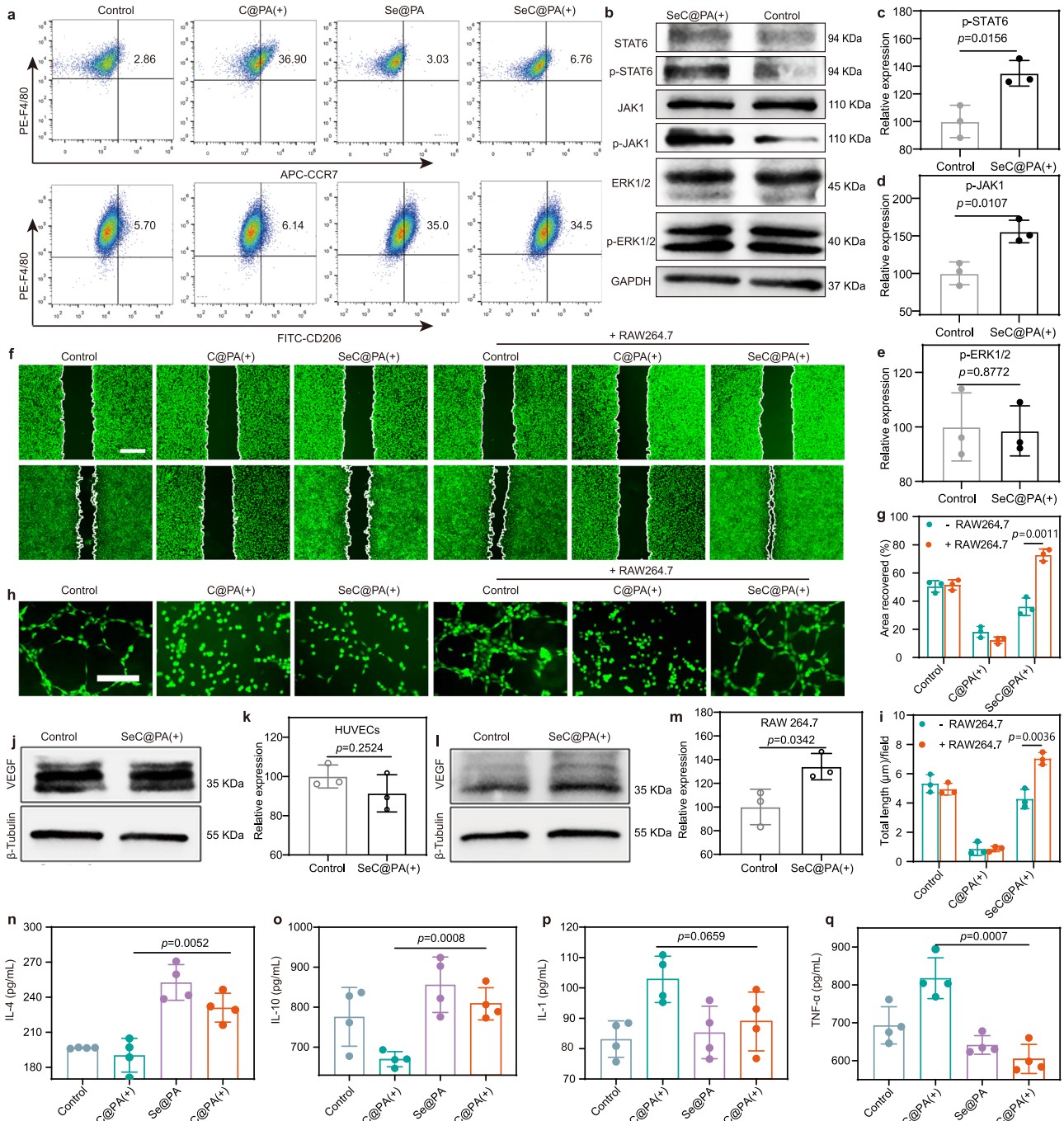

**Fig. 5 | SeC@PA effectively polarized macrophages toward M2 phenotype.**
**a** Flow cytometry analysis of F4/80⁺ CCR7⁺ and F4/80⁺ CD206⁺ cells. **b–e** Expression levels of p-STAT6, p-JAK1 and p-ERK1/2 in the RAW264.7 cells determined by western blotting after treatments ($n = 3$ biologically independent samples; mean ± SD). **f** Representative images of migration of HUVECs treated with different groups. Scale bar is 500 μm. **g, i** Quantitative analysis of migration and tube formation of HUVECs, respectively ($n = 3$ biologically independent samples; mean ± SD). **h** Representative images of tube formation of HUVECs treated with different groups. Scale bar is 200 μm. **j–m** Expression levels of VEGF in the HUVECs and

RAW264.7 cells determined by western blotting ($n = 3$ biologically independent samples; mean ± SD). **n–q** Inflammatory cytokine levels of different group ($n = 4$ biologically independent samples; mean ± SD). The experiments in **b, j, l** were repeated three times with similar results. Statistical significance was analyzed via two-tailed Student's $t$ test (**c–e, g, i, k,** and **m**) or one-way ANOVA with a Tukey post-hoc test (**n–q**). Source data are provided as a Source Data file. Se@PA: Se-PDA-LA nanoparticles; C@PA(+): Ce6-PDA-LA nanoparticles under 660 nm irradiation (0.2 W/cm²) for 3 min; SeC@PA(+): Se-Ce6-PDA-LA nanoparticles under 660 nm irradiation (0.2 W/cm²) for 3 min.

These results indicated that RS produced by PDT treatment alone (C@PA MN(+)) has the risk of delaying diabetic chronic wound healing. It is necessary to remove the RS produced by PDT immediately to avoid secondary injury to the wound tissues. SeC@PA MN (+) could effectively remove the RS produced by PDT in wound tissues, alleviate oxidative stress, and promote wound healing.

## SeC@PA MN for promoting biofilm-infected chronic wound healing by efficient delivery of SeC@PA

Chronic diabetic wounds are mostly covered by inflammatory exudates and biofilms (Supplementary Fig. 26), forming physicochemical barriers that greatly reduce the bioavailability of topical drugs. To demonstrate the advantages of microneedle bandage in the treatment

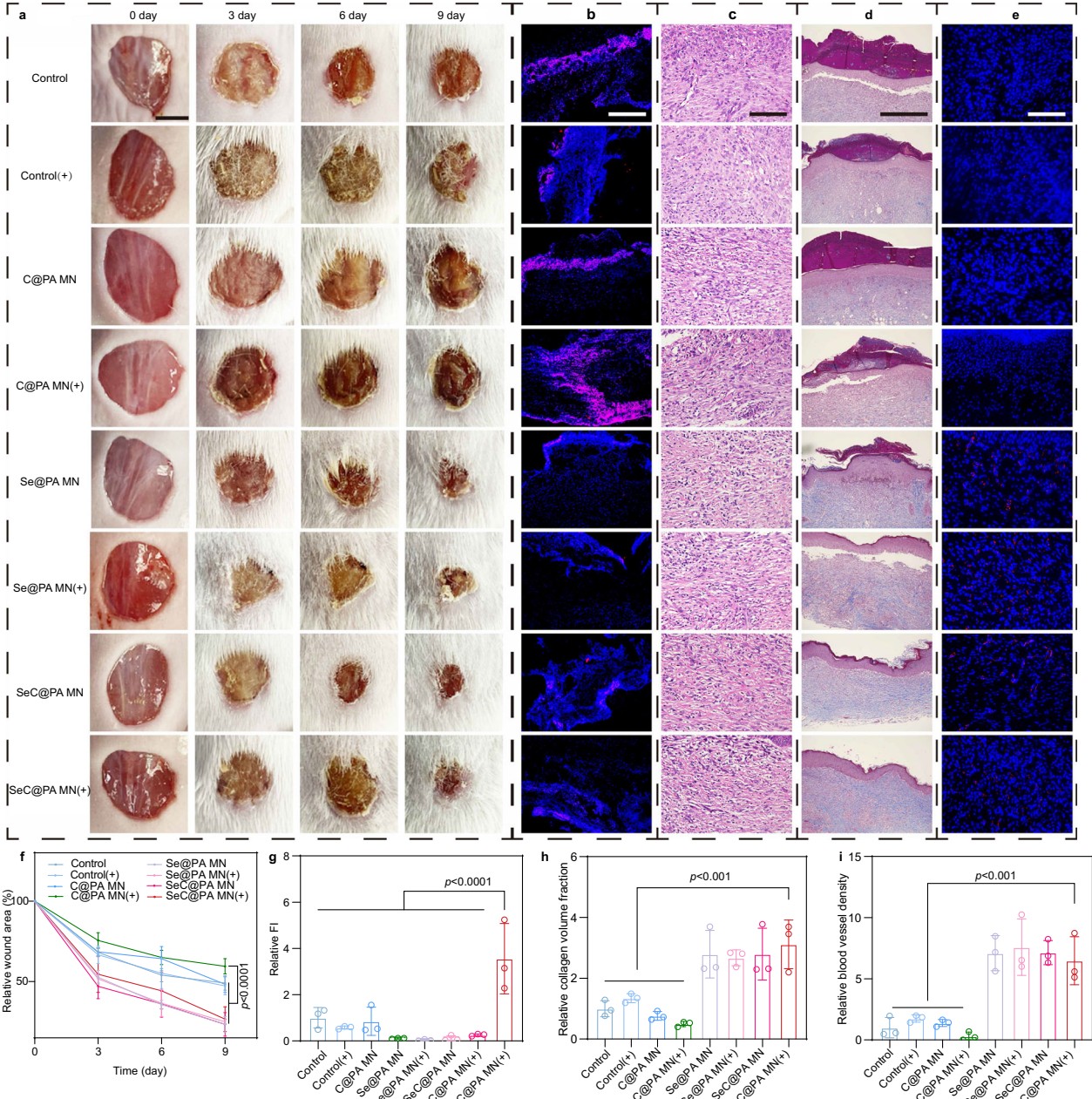

**Fig. 6 | SeC@PA MN promoted the healing of non-biofilm-infected full-thickness diabetic wounds in rats. a, f** Representative images of wounds during healing and quantitative data of relative wound area to day 9 of the different groups at different time points ($n = 5$ biologically independent samples; mean ± SD). Scale bar is 5 mm. **b, g** Representative immunofluorescence images of RS staining and relative fluorescence intensity of RS at day 2 ($n = 3$ biologically independent samples; mean ± SD). Scale bar is 100 μm. **c** Representative H&E staining images of wound samples treated with different groups on day 9. Scale bar is 100 μm. Three independent experiments were performed and representative results are shown. **d, h** Representative Masson's trichrome-stained images of wound samples and quantification of the collagen volume fraction for different groups on day 9 ($n = 3$ biologically independent samples; mean ± SD). Scale bar is 500 μm. **e, i** Representative immunofluorescence images of CD31 staining in the

regenerated skin tissue and quantification of blood vessel density on day 9 after wound healing ($n = 3$ biologically independent samples; mean ± SD). Scale bar is 100 μm. Statistical significance was analyzed via one-way ANOVA with a Tukey post-hoc test. Source data are provided as a Source Data file. Control: blank microneedle; Control(+): blank microneedle under 660 nm irradiation ($0.2$ W/cm²) for 3 min; C@PA MN: microneedle containing Ce6-PDA-LA nanoparticles; C@PA MN(+): microneedle containing Ce6-PDA-LA nanoparticles under 660 nm irradiation ($0.2$ W/cm²) for 3 min; Se@PA MN: microneedle containing Se-PDA-LA nanoparticles; Se@PA MN(+): microneedle containing Se-PDA-LA nanoparticles under 660 nm irradiation ($0.2$ W/cm²) for 3 min; SeC@PA MN: microneedle containing Se-Ce6-PDA-LA nanoparticles; SeC@PA MN(+): microneedle containing Se-Ce6-PDA-LA nanoparticles under 660 nm irradiation ($0.2$ W/cm²) for 3 min.

of diabetic chronic wounds, we compared the therapeutic effects of SeC@PA MN(+) with SeC@PA NPs(+). As shown in Fig. 7, the SeC@PA MN(+) group showed a strong impact in promoting wound healing (Fig. 7a, g) with higher collagen deposition fraction (Fig. 7c, h) and vessel density (Fig. 7d, i) relative to the control group and SeC@PA

NPs(+) group. However, there was no obvious difference between the SeC@PA NPs(+) group and the control group in promoting wound healing.

To further explore whether the location of SeC@PA in the wound is responsible for the differences in therapeutic effects between

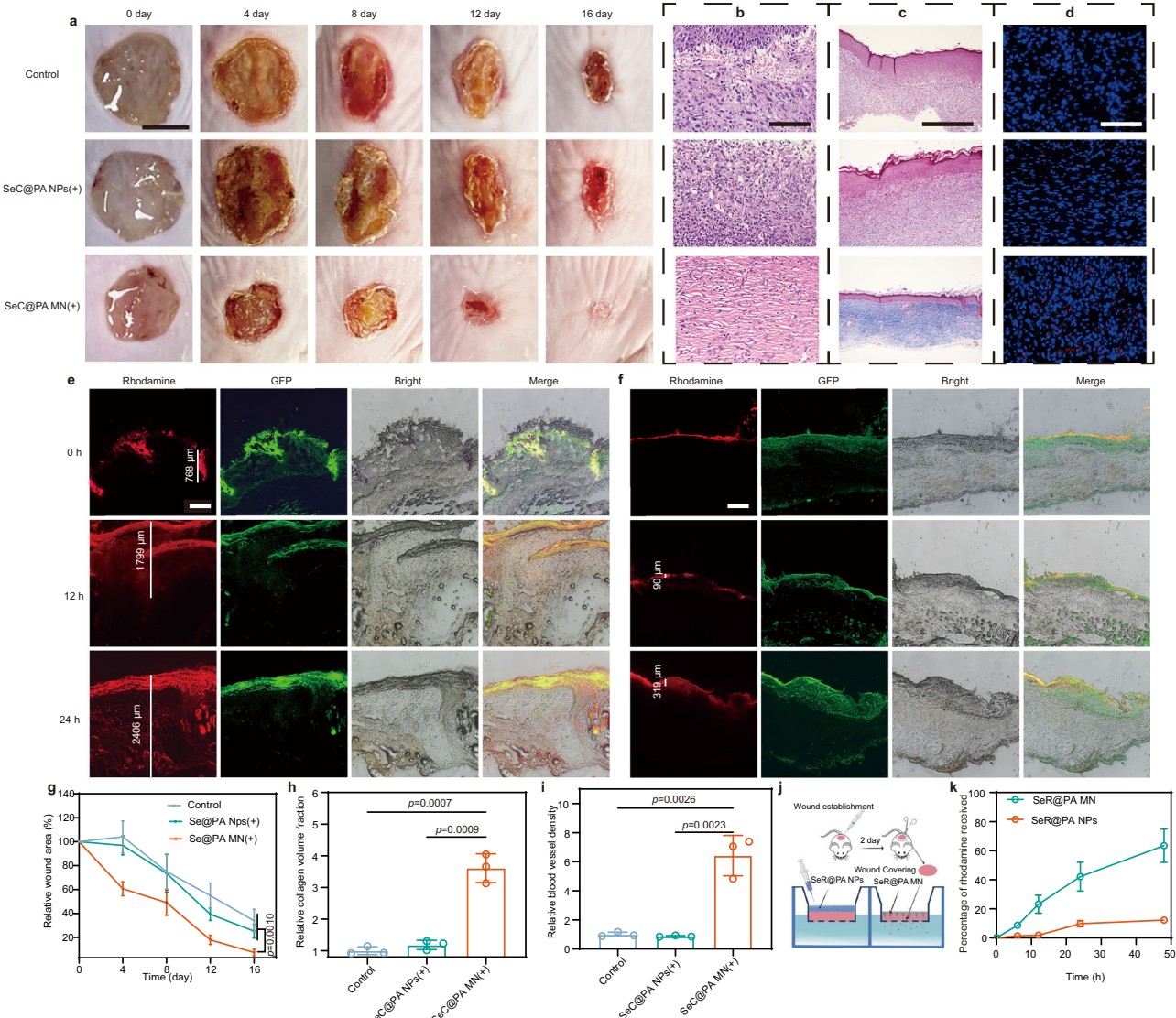

**Fig. 7 | SeC@PA MN for promoting chronic wound healing through efficient delivery of SeC@PA. a**, **g** Representative images of wounds during healing and quantitative data of relative wound area to day 16 of the different groups at different time points (*n* = 4 biologically independent samples; mean ± SD). Scale bar is 5 mm. **b** Representative H&E staining images of wound samples treated with different groups on day 16. Scale bar is 100 μm. Three independent experiments were performed and representative results are shown. **c**, **h** Representative Masson's trichrome-stained images of wound samples and quantification of the collagen volume fraction for three groups on day 16 (*n* = 3 biologically independent samples; mean ± SD). Scale bar is 500 μm. **d**, **i** Representative immunofluorescence images of CD31 staining in the regenerated skin tissue and quantification of blood vessel density on day 16 after wound healing (*n* = 3 biologically independent samples; mean ± SD). Scale bar is 100 μm. **e** Representative images of distribution of

SeR@PA in wound coverings at different time in SeR@PA MN group. Scale bar is 500 μm. **f** Representative images of distribution of SeR@PA in wound coverings at different time in SeR@PA NPs group. Scale bar is 500 μm. **j** Schematic representation for the acquisition of ex vivo wound coverages and determination of penetration efficiency of SeR@PA in wound coverings. **k** Penetration efficiency of SeR@PA MN and SeR@PA NPs in wound coverings (*n* = 3 biologically independent samples; mean ± SD). Statistical significance was analyzed via one-way ANOVA with a Tukey post-hoc test. Source data are provided as a Source Data file. SeC@PA NPs(+): solution containing Se-Ce6-PDA-LA nanoparticles under 660 nm irradiatio (0.2 W/cm²) for 3 min; SeC@PA MN(+): microneedle containing Se-Ce6-PDA-LA nanoparticles under 660 nm irradiation (0.2 W/cm²) for 3 min; SeR@PA NPs: solution containing Se-Rhodamine-PDA-LA nanoparticles; SeR@PA MN: microneedle containing Se-Rhodamine-PDA-LA nanoparticles.

SeC@PA MN(+) group and SeC@PA NPs(+) group, we investigated the distribution of SeC@PA after administration. Firstly, biofilms were generated by infecting chronic diabetic wounds with *E. coli* expressing GFP for easy observation. Secondly, SeR@PA MN and SeR@PA NPs were prepared by replacing Ce6 with rhodamine for better observation of the distribution of SeC@PA in the wound coverings. After two days of biofilm growth, SeR@PA MN and SeR@PA NPs were applied to the wound, respectively. As shown in Fig. 7e, f, SeR@PA MN effectively punctured the wound coverings and delivered SeR@PA to the underside of the wound coverings, and the penetration depth of SeR@PA was increased with the duration of administration. However,

SeR@PA from SeR@PA NPs was mostly captured by the biofilms on the surface of the wound coverings, which could reach neither the biofilms deep in the wound coverings nor the area below it. To investigate the penetration efficiency of SeR@PA from SeR@PA MN and SeR@PA NPs into the wound coverings, the wound coverings were placed into transwells, which were then inserted into a 24-well plate to determine the amount of SeR@PA permeated into the well plates (Fig. 7j). The SeR@PA MN group showed higher wound covering penetration efficiency compared to that of SeR@PA NPs (Fig. 7k). At the same time, we performed bacterial coated plate counting on the wounds after 4 days of treatments with SeC@PA MN(+) and SeC@PA NPs(+), showing that

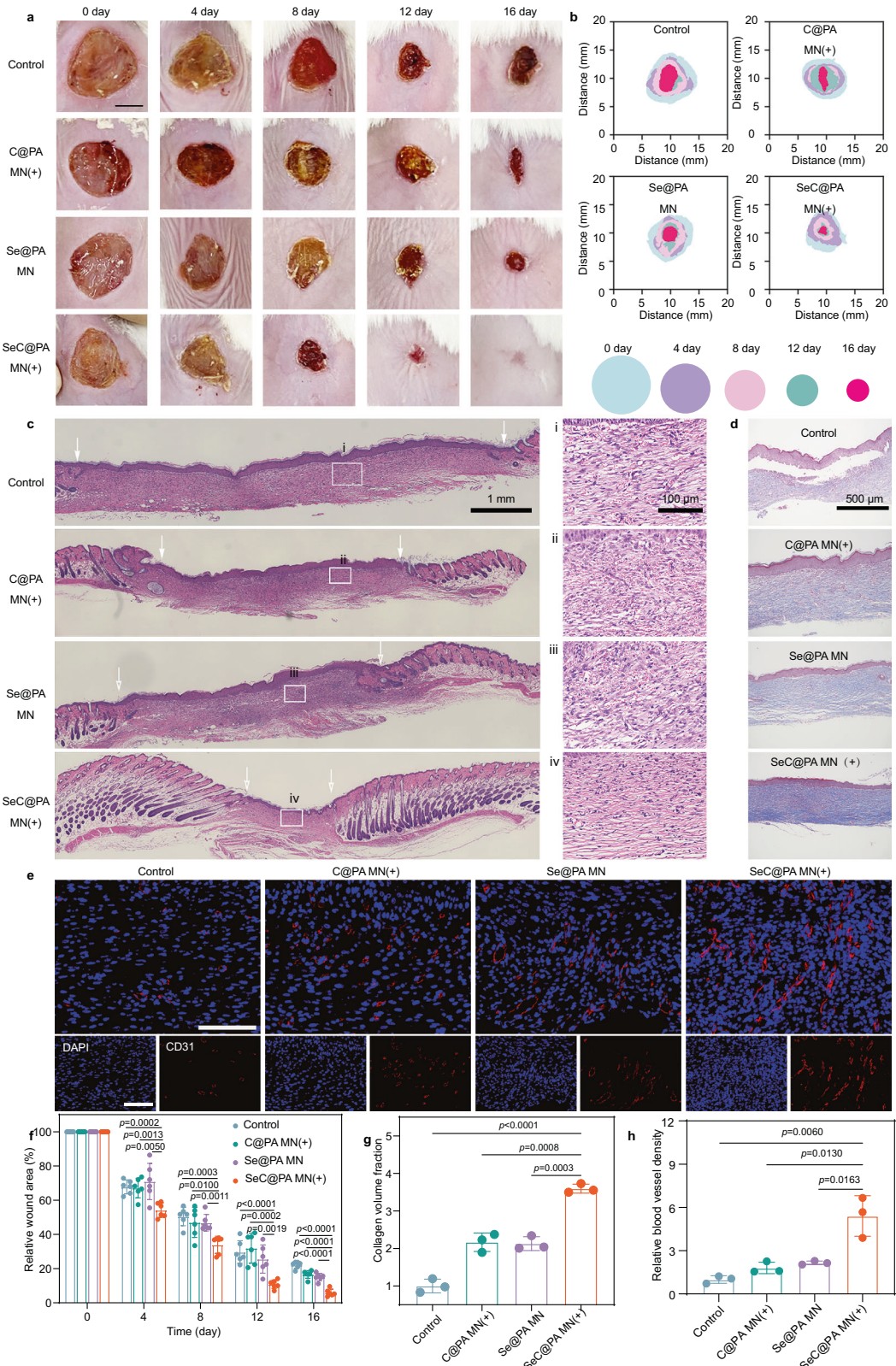

the antibacterial effect of SeC@PA MN(+) was much stronger than that of SeC@PA NPs (+) group (Supplementary Fig. 27).

These results demonstrated the advantages exhibited by MNs in the field of treating chronic wounds. Chronic diabetic wounds are often covered with inflammatory exudates and biofilms, which form a physicochemical barrier that prevents the penetration of therapeutic drugs into the action sites beneath the wound

coverings. Therefore, conventional local delivery methods have limitations in the treatment of chronic wounds. On the contrary, MNs can penetrate the physicochemical barrier on the wound surface to achieve deeper and wider drug delivery, effectively eradicating the biofilm, reducing oxidative stress, and promoting angiogenesis and collagen deposition in the wound to facilitate wound healing.

**Fig. 8 | SeC@PA MN for promoting the healing of biofilm-infected full-thickness diabetic wounds in rats. a** Representative images of wounds during healing. Scale bar is 5 mm. **b** Schematic diagram of the wound-healing process of the four groups. **c** Representative H&E staining images of wound samples treated with different groups on day 16. Scale bar is 1 mm. Three independent experiments were performed and representative results are shown. **d, g** Representative Masson's trichrome-stained images of wound samples and quantification of the collagen volume fraction for four groups on day 16 ($n = 3$ biologically independent samples; mean ± SD). Scale bar is 500 μm. **e, h** Representative immunofluorescence images of CD31 staining in the regenerated skin tissues and quantification of blood vessel density on day 16 after wound treatment ($n = 3$ biologically independent samples; mean ± SD). Scale bar is 100 μm. **f** Quantitative data of relative wound area to day 16 of the four groups at different time points ($n = 6$ biologically independent samples; mean ± SD). Statistical significance was analyzed via one-way ANOVA with a Tukey post-hoc test. Source data are provided as a Source Data file. Control: blank microneedle; C@PA MN(+): microneedle containing Ce6-PDA-LA nanoparticles under 660 nm irradiation (0.2 W/cm²) for 3 min; Se@PA MN: microneedle containing Se-PDA-LA nanoparticles; SeC@PA MN(+): microneedle containing Se-Ce6-PDA-LA nanoparticles under 660 nm irradiation (0.2 W/cm²) for 3 min.

## Treatment of biofilm-infected wounds in diabetic mice using SeC@PA MN bandage

Subsequently, we investigated the effects of SeC@PA MN on biofilm-infected diabetic chronic wounds. Firstly, we determined GSH content in wound coverings (including biofilms and inflammatory exudates) and wound tissues (beneath the wound coverings) to determine whether there were differences in GSH content to achieve the bidirectional regulatory effect of SeC@PA on RS. As displayed in Supplementary Fig. 28, the GSH content in the wound coverings was much higher than that in the wound tissue, with a content as high as 6.69 mmol/g, which was 9.3 times higher than 0.72 mmol/g in the tissue. This difference in GSH content could provide a basis for achieving different regulatory effects of SeC@PA on RS at the two sites.

SeC@PA MN significantly promoted wound healing after laser irradiation (Fig. 8 and Supplementary Fig. 29). After 16 days of treatment, wounds in the SeC@PA MN(+) group were almost healed with a healing rate >95%, whereas 22%, 16%, and 15% of the wound areas remained in the control group, C@PA MN(+) group, and Se@PA MN group, respectively (Fig. 8a, b, f). Hematoxylin-eosin (HE) staining (Fig. 8c) and Masson's trichrome staining (Fig. 8d) were performed to investigate the histological changes of wounds treated with different treatment. On day 16, the control group showed only a few collagen deposits (Fig. 8g) and irregular skin structure, which was infiltrated by many inflammatory cells. Similar phenomena were found in the C@PA MN(+) and Se@PA MN groups. In the SeC@PA MN(+)-treated wounds, the epidermis and dermis of the skin were relatively intact and highly ordered, and inflammatory cell infiltration was less. Additionally, the relative CD31 fluorescence intensity (Fig. 8e, h and Supplementary Fig. 30) increased remarkably in SeC@PA MN(+)-treated wounds, suggesting a significantly higher density of vessels in the dermis from wounds cured by SeC@PA MN(+) compared with those in other groups.

Dihydroethidium staining was used to determine RS levels in wound tissues after different treatments on day 4. The experimental procedures are shown in Fig. 9a. The control group and C@PA MN(+) group displayed strong red fluorescence, indicating that the oxidative stress in wound tissue was severe (Fig. 9b, f and Supplementary Fig. 31). After the application of Se@PA MN and SeC@PA MN(+), RS levels decreased obviously, demonstrating that both MNs could scavenge RS. At the same time, we determined GPX4 level and activity in wound tissues (Fig. 9e, h, i) and found that SeC@PA MN could enhance both parameters. Unexpectedly, Se@PA MN did not obviously increase GPX4 activity and level in vivo, which was different from the results from in vitro studies. Moreover, the number of colonies in wound coverings after SeC@PA MN(+) treatment was much lower compared with other treatments (Fig. 9c, d, g), demonstrating the excellent anti-biofilm ability of SeC@PA MN(+) in vivo.

It is worth noting that the healing of chronic diabetic wounds is a complex process that requires simultaneous anti-biofilm and anti-inflammatory effects to promote healing. Se@PA MN can scavenge RS but cannot eradicate biofilms, exacerbating the inflammatory response in wounds. C@PA MN(+) can kill most bacteria by generating RS, which further aggravates the inflammatory response of wounds. Therefore, neither Se@PA MN nor C@PA MN(+) can effectively

promote wound healing, which was consistent with the results of wound healing in our study. In addition, the detection of RS in Se@PA was limited to the early stage of wound healing (at day 4). Persistent bacterial infection aggravates the oxidative stress response in wounds and inactivates the functions of some enzymes such as GPX4, which may explain why Se@PA MN did not significantly increase GPX4 activity and levels. Therefore, SeC@PA MN(+) effectively promoted wound healing by eradicating biofilms and reducing inflammation, thereby increasing angiogenesis and collagen deposition.

## Macrophage polarization by SeC@PA MN in vivo

An important indicator of the inflammatory response is the phenotype of macrophages[44]. It is well known that the infiltration of chronic wounds by M1-like macrophages results in a persistent inflammatory response that hinders wound healing. Therefore, modulation of macrophage polarization to the M2-phenotype with anti-inflammatory activity can benefit wound closure. To further illustrate the effect of SeC@PA MN(+) on macrophage polarization, the infiltration of macrophages into wounds was investigated using immunofluorescence staining. The wounds treated by SeC@PA MN(+) exhibited decreased M1-phenotype macrophage (F4/80⁺ and CD86⁺) infiltration and increased M2-phenotype macrophage (F4/80⁺ and CD206⁺) infiltration as compared with the control group and C@PA MN(+) group (Fig. 10a–d, Supplementary Fig. 32 and Supplementary Fig. 33). This result is attributed to Se promoting macrophage polarization to the M2-phenotype. In addition, SeC@PA MN(+) significantly increased the levels of the anti-inflammatory factors IL-4 and IL-10 and decreased those of the pro-inflammatory factors IL-1 and TNF-α, further demonstrating that macrophages were polarized to the M2- phenotype (Fig. 10e–h). Overall, SeC@PA MN(+) could promote the polarization of macrophages into an anti-inflammatory M2-phenotype and promote wound healing.

## Discussion

Infected biofilms, persistent inflammation, and impaired angiogenesis are the hallmarks of diabetic chronic wounds. There is an urgent need to develop anti-biofilm, anti-inflammatory, and pro-angiogenic multifunctional agents to promote the healing of chronic diabetic wounds. PDT has been widely studied as antibacterial therapy owing to its virtue of generating ROS. However, high GSH concentrations in biofilms limit the therapeutic effect of PDT. In addition, ROS plays a dual role in eradicating biofilms and exacerbating inflammation. PDT produces a large amount of ROS in a short time. Although these ROS are transient, they can interfere with cell function, including that of HUVECs, and delay wound healing, especially in chronic wounds with severe oxidative stress, where this damage is more severe. Resolving this issue necessitates the introduction of additional ROS scavengers, which not only complicates the system but also impedes its clinical translation. Moreover, a frequently overlooked challenge lies in selecting the appropriate site of action, particularly in the context of chronic wound treatment. Chronic wounds are often covered with inflammatory exudates, which, together with biofilms, form barriers. Topically used drugs must pass through the barrier to reach the underlying live cells while combating the exudate transport occurring in the reverse

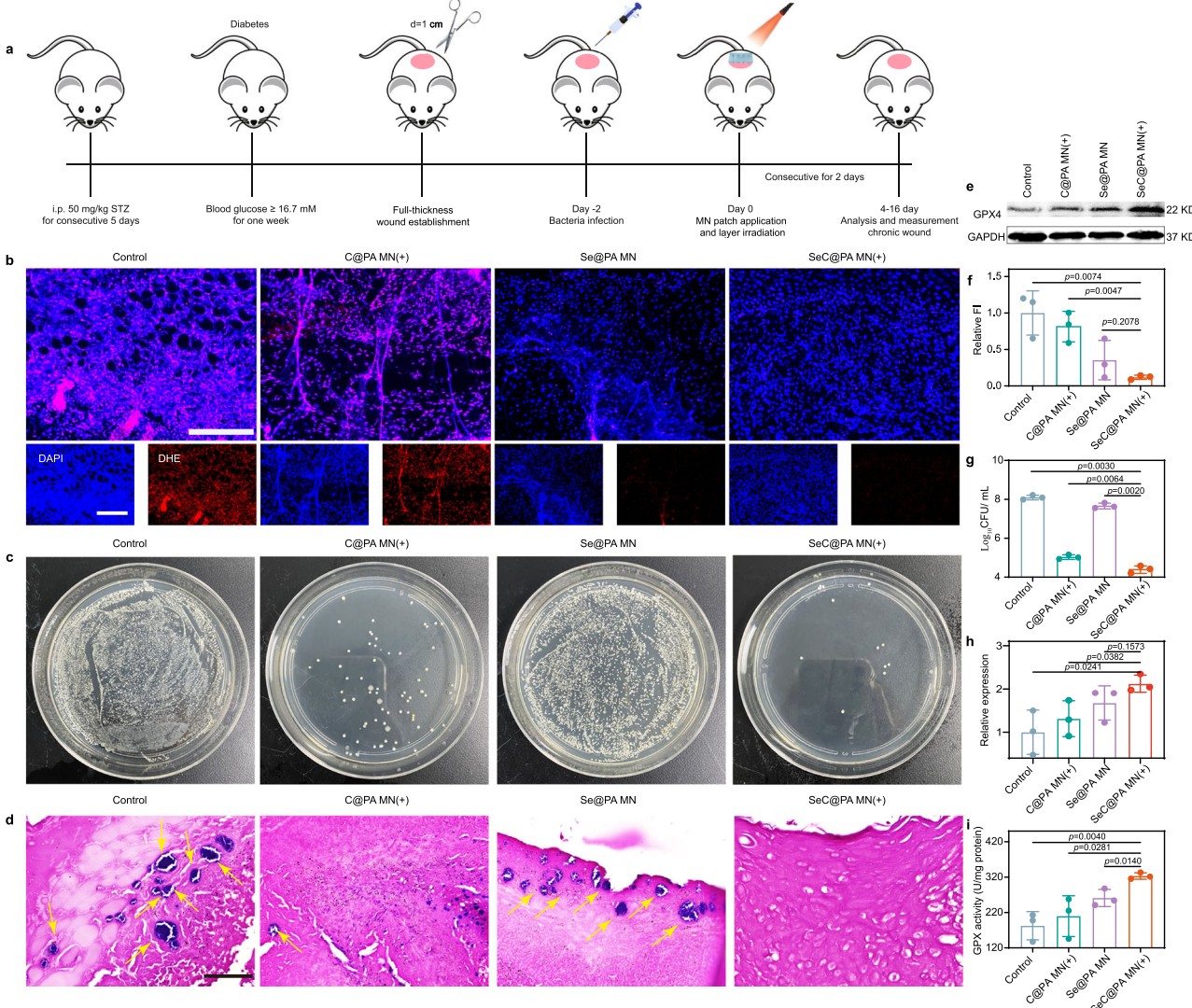

**Fig. 9 | SeC@PA MN for reducing wound inflammation and bacteria to promote angiogenesis. a** Schematic illustration of the experimental procedure for treating biofilms-infected mice with diabetic. **b, f** Representative immunofluorescence images of RS staining and relative fluorescence intensity of RS at day 4 ($n = 3$ biologically independent samples; mean ± SD). Scale bar is 100 μm.
**c, g** Representative photographs of bacterial culture from the skin tissue of diabetic mouse wounds infected with biofilms and bactericidal results at day 4 characterized by the standard plate counting assay ($n = 3$ biologically independent samples; mean ± SD). **d** Gram-staining images of the skin tissue of diabetic mice wounds infected with biofilms at day 4. The yellow arrows represent bacteria. Scale bar is 50 μm. Three independent experiments were performed and representative results

are shown. **e, h** Expression levels of GPX4 from the wound tissues of diabetic mice infected with biofilms at day 8 determined by western blotting ($n = 3$ biologically independent samples; mean ± SD). The experiment in **e** was repeated three times with similar results. **i** Activity of GPX4 from the wound tissue of diabetic mice infected with biofilms at day 8 ($n = 3$ biologically independent samples; mean ± SD). Statistical significance was analyzed via one-way ANOVA with a Tukey post-hoc test. Source data are provided as a Source Data file. Control: blank microneedle; C@PA MN(+): microneedle containing Ce6-PDA-LA nanoparticles under 660 nm irradiation (0.2 W/cm²) for 3 min; Se@PA MN: microneedle containing Se-PDA-LA nanoparticles; SeC@PA MN(+): microneedle containing Se-Ce6-PDA-LA nanoparticles under 660 nm irradiation (0.2 W/cm²) for 3 min.

direction, and therefore, the bioavailability of topical drugs is greatly reduced.

In this study, we design a single and highly effective SeC@ PA MN bandage to treat chronic diabetic wounds. As an emerging delivery system, MNs have been widely studied for their suitability in wound healing and other biological applications[30,45–53]. In our work, MN is used to puncture the physicochemical barrier covering the wound bed, leading to the subsequent delivery of cargo to the biofilms and active tissue. Compared with the local delivery SeC@PA NPs, the SeC@PA MN bandage can deliver SeC@PA directly to the biofilms and wound tissues below the wound coverings, greatly improving the bioavailability of the drug. When SeC@PA is delivered to biofilms with high GSH levels, •OH, which is more toxic than ¹O₂ (mainly produced by Ce6), is generated during

irradiation through a series of cascade decomposition reactions. Simultaneously, NO, produced by LA, is employed to deplete GSH and generate RNS, forming an RS storm to eradicate the biofilm. This therapeutic effect cannot be achieved merely by depleting GSH to assist PDT. Both in vitro and in vivo studies demonstrate that the bacteria in the biofilms are efficiently eliminated by SeC@PA(+), indicating the effectiveness of the self-enhancing anti-biofilm strategy at high GSH levels. When applied to wound tissues with low GSH levels, SeC@PA acts as an RS scavenger, quickly quenches the RS produced by SeC@PA after irradiation, and increases the levels and activity of GPX4, thereby reducing wound inflammation. These results suggest that SeC@PA could dynamically regulate RS without extra additives by sensing different GSH levels, simultaneously eradicating biofilms and inhibiting inflammation.

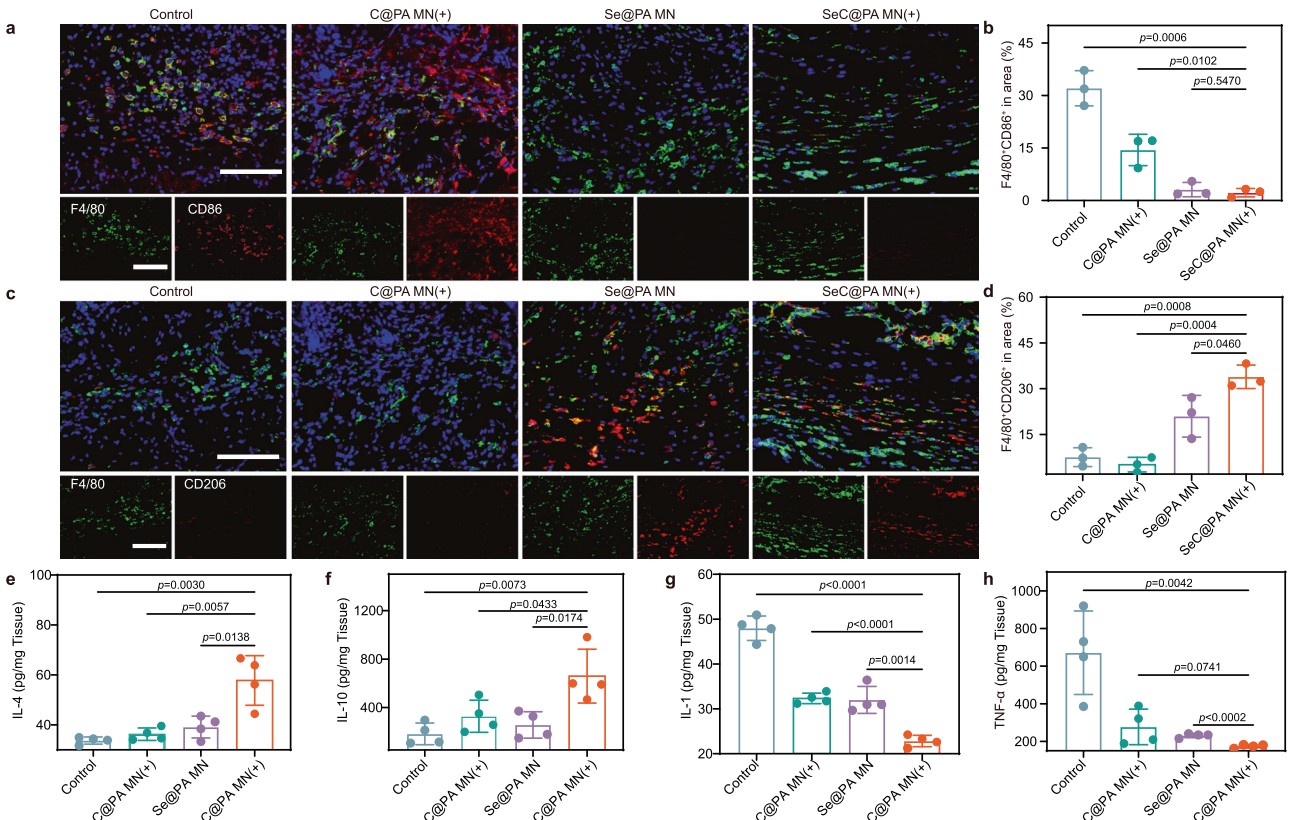

**Fig. 10 | SeC@PA MN for relieving inflammation and stimulating M2 polarization of macrophages in chronic diabetic wounds. a** Immunofluorescence images of F4/80 (green) and CD86 (red) in the infected tissue on day 8. Scale bar is 100 μm. Three independent experiments were performed and representative results are shown. **b, d** Percentage of F4/80$^+$ CD86$^+$ cells and F4/80$^+$ CD206$^+$ cells in area ($n = 3$ biologically independent samples; mean ± SD). **c** Immunofluorescence images of F4/80 (green) and CD206 (red) in the infected tissue at day 8. Scale bar is 100 μm. Three independent experiments were performed and representative results are shown. **e–h** Inflammatory cytokine levels of biofilms infected diabetic wounds at day 8 ($n = 4$ biologically independent samples; mean ± SD). Statistical significance was analyzed via one-way ANOVA with a Tukey post-hoc test. Source data are provided as a Source Data file. Control: blank microneedle; C@PA MN(+): microneedle containing Ce6-PDA-LA nanoparticles under 660 nm irradiation (0.2 W/cm$^2$) for 3 min; Se@PA MN: microneedle containing Se-PDA-LA nanoparticles; SeC@PA MN(+): microneedle containing Se-Ce6-PDA-LA nanoparticles under 660 nm irradiation (0.2 W/cm$^2$) for 3 min.

Another obstacle in the healing process of chronic diabetic wounds is the infiltration of M1-phenotype macrophages. The anti-inflammatory effect due to the scavenging of ROS is transient. Remodeling the immune environment of the wound beds, especially polarizing macrophages to the M2-phenotype with a sustained anti-inflammatory effect, is necessary to promote healing. In this study, the SeC@PA clearly improves macrophage polarization into the M2-phenotype through the JAK-STAT6 pathways, promoting the levels of the anti-inflammatory factors IL-4 and IL-10 and reducing the levels of the pro-inflammatory factors. Macrophage polarization to the M2-phenotype promotes HUVEC migration and tube formation in vitro by secreting VEGF. The infiltration of M2-phenotype macrophages in wounds obviously increases in diabetic mice with chronic wounds, facilitating the inhibition of inflammation and promoting angiogenesis. Thus, this study represents the initial exploration of a potential therapeutic approach for chronic wounds by targeting the distinct microenvironments of biofilms and wound beds. These findings offer valuable insights into the rational design of topical treatments for the effective management of chronic wounds.

## Methods

### Preparation of Se-Ce6-DA-LA nanoparticles (SeC@PA)

Na$_2$SeO$_3$•5H$_2$O (26.3 mg) was dissolved in ultrapure water (10 mL) containing 1% (v/v) tween 20. Next, vitamin C (70 mg) was dissolved in ultrapure water (10 mL). Na$_2$SeO$_3$ solution was added to vitamin C solution and stirred at room temperature for 10 h, and Se

nanoparticles were collected by centrifugation. Subsequently, Se (1.5 mg) was suspended in ultrapure water (1 mL), followed by the addition of Ce6 (0.3 mg) and DA (10 mg) and adjustment of the pH with NaOH to 9.0. The mixture was stirred at room temperature for 24 h to obtain Se-Ce6-DA nanoparticles (SeC@P). Lastly, LA (50 mg) was added to SeC@P solution under stirring at room temperature for 12 h to obtain Se-Ce6-DA-LA nanoparticles (SeC@PA). The hydrodynamic size and zeta potential of Se, SeC@P, and SeC@PA were measured using a zeta potential analyzer (Malvern Mastersizer 2000, UK). The levels of Se, Ce6, and LA in SeC@PA were determined using inductively coupled plasma atomic emission spectrometry (PerkinElmer Optima8300, USA), ultraviolet spectrophotometry (Tecan/Infinite E plex, Switzerland), and the Sakaguchi reaction, respectively. The TEM (FEI Tecnai G2 F30, Netherlands) and scanning electron microscope (SEM, Regulus 8230, Japan) were employed to observe the appearance and detect the elemental composition of the SeC@PA, respectively.

The preparation of Ce6-DA-LA nanoparticles (C@PA) was similar to SeC@PA but without the addition of Se. The preparation of Se-DA-LA nanoparticles (Se@PA) was similar to SeC@PA but without the addition of Ce6.

### Preparation of SeC@PA MN bandage

SeC@PA MNs were prepared using the centrifugal infusion poly-dimethylsiloxane (PDMS) mold method[54,55]. Briefly, SeC@PA was suspended in an aqueous solution containing 10% sucrose and 18% collagen, and then the solution (200 μL) was added to the PDMS

mold using a pipette, followed by centrifugation (Centrifuge 5910R, Eppendorf, Shanghai, China) at $4600 \times g$ and 4 °C for 3 min to fill the cavities. The residual suspension was removed with a pipette. Subsequently, the mold was centrifuged at 25 °C for 30 min to dry. Then, 18% aqueous polyvinyl alcohol (PVA) solution was added to the mold, centrifuged (Eppendorf) for 10 min ($4600 \times g$, 4 °C), and stored in a desiccator overnight to obtain SeC@PA MNs. Se, Ce6, and LA levels in each MN patch were approximately 158.0 µg, 64.7 µg, and 97.1 µg, respectively. A biofilm was overlaid on an agarose gel to simulate the barriers of chronic wounds and the punctures of the barriers by the MNs were evaluated.

### Detection of NO
NO levels were measured using a NO assay kit (Beyotime Biotechnology, Shanghai). Briefly, different concentrations of SeC@PA were exposed to a 660 nm laser (0.2 W/cm²) for different time periods, or SeC@PA was exposed to a 660 nm laser at different irradiation intensities. A NO standard curve was used to determine NO levels based on the absorbance of the sample at 550 nm.

### Degradation of GSH in vitro
SeC@PA and C@PA were dispersed in GSH aqueous solution (8 mM). After exposure to a 660 nm laser for different time intervals, the GSH level remaining in the solution was determined using a microreduced GSH assay kit (Solarbio Science & Technology Co., Ltd., Beijing).

### Extracellular measurement of RS
DCF-DA was used to determine the generation of total RS. Briefly, C@PA or SeC@PA (1 mg) was dispersed in PBS (1 mL, pH = 5.5) with or without GSH or with different concentrations of GSH. DCF-DA was added and irradiated using a 660 nm laser (0.2 W/cm², 3 min). Subsequently, the fluorescence intensity was determined using an enzyme-linked immunosorbent assay (ELISA) analyzer (Tecan/Infinite E plex, Switzerland). EPR technique (EPR, Bruker EMXplus, Germany) was used to quantify the generation of •OH with DMPO. Different groups were treated with GSH, without GSH, or with different concentrations of GSH. Then, DMPO (60 µL) was added to determine •OH generation at room temperature with irradiation. MB was also used to quantify •OH generation. Thus, different groups were or were not treated with GSH. After irradiation for 3 min, the generation of •OH was determined by measuring the absorbance at 664 nm using a microplate reader.

### Culturing PA and SA biofilms
*Staphylococcus aureus* (SA, ATCC 25923) and *Pseudomonas aeruginosa* (PA, ATCC 27853) were used in this study. For biofilm culturing, an SA suspension (10 µL, $10^8$ CFU/mL) and Mueller-Hinton broth medium (MHB, 300 µL) were placed in 48-well plates and cultured at 37 °C for 2 d. The pH of the biofilm was adjusted by adding PBS (pH = 5.5) for in vitro experiments.

### In vitro anti-biofilm effect of nanoparticles
The anti-biofilm effect of different nanoparticles was studied using live-dead staining, plate-counting assays, and crystal violet staining. For live-dead staining, different nanoparticles were incubated with the biofilms for 4 h and irradiated for 3 min. Two hours later, the treated biofilms were stained using a bacterial live-dead staining kit. Live and dead bacteria were observed using confocal laser microscopy (Zeiss LSM 880, Germany). For plate-counting assays, the treated biofilms were collected into PBS and dissolved by violent vortex. The number of bacteria was counted using the dilution-plate counting method. After treatment, 0.5% crystal violet solution was added to the biofilms, stained for 30 min, and washed 6 times with PBS. Lastly, the crystal violet was dissolved in methanol, and the absorbance at 590 nm was measured using a microplate reader. After treatment, the RS in

biofilms was detected using DCF-DA and observed using fluorescence microscopy (ZEISS Axio Observer 3, Germany). GSH levels were determined using a micro-reduced GSH assay kit.

### In vitro anti-inflammatory effect of nanoparticles
Different nanoparticles and HUVECs were incubated for 6 h and exposed to irradiation at 660 nm (0.2 W/cm²) for 3 min. RS production was determined using DCF-DA and observed using fluorescence microscopy (ZEISS Axio Observer 3, Germany). The generated RS was quantitatively analyzed using flow cytometry. Immunofluorescence staining of GPX4 was also performed. Western blot was used to determine GPX4 levels, and GPX activity was measured using a GSH peroxidase (GSH-Px/GPX) activity detection kit (Solarbio Science & Technology Co., Ltd, Beijing).

### Macrophage polarization assessment in vitro
C@PA, Se@PA, SeC@PA, and RAW264.7 cells were incubated for 6 h and subsequently exposed to a 660 nm laser for 3 min. Total macrophages were labeled with rabbit F4/80 polyclonal primary antibody (M1 macrophages were labeled with rabbit CCR7 polyclonal primary antibody, and M2 macrophages were labeled with rabbit CD206 polyclonal primary antibody). The samples were subsequently analyzed using flow cytometry (Beckman cytoflex, Germany). qRT-PCR (ABI QuantStudio 5, USA) was used to determine the expression of specific genes in RAW264.7 macrophages after induction by nanoparticles (Supplementary Table 2). Western blotting was used to elucidate the mechanism of RAW264.7 cell polarization. The detailed procedure is provided in the Supplementary Information.

### Assessment of cell migration and tube formation in vitro
After culturing HUVECs, scratches were created using a 200-µL pipette. C@PA and SeC@PA were added to cells and co-cultured for 6 h before laser irradiation for 3 min. For the group with RAW264.7, a transwell with a pore size of 0.4 µm coated with RAW264.7 cells was placed into the well plate containing HUVECs (Supplementary Fig. 25a). Wound healing was observed using fluorescence microscopy at 0 and 12 h after treatment. Tube formation was assessed using a six-well plate precoated with Matrigel matrix (BD, Corning, US) (Supplementary Fig. 25b). Western blotting was used to analyze the changes in VEGF in different treatment groups.

### In vivo assessment of diabetic wound healing
Male BALB/c were purchased from the Laboratory Animal Center of Sun Yat-Sen University. All protocols for in vivo experiments were approved by the Institutional Animal Care and Use Committee of Sun Yat-Sen University (SYSU-IACUC-2022-000945). All the mice (five mice per cage) were housed in standard, infection-free housing room, with 12 h light:12 h dark cycles in the vivarium. Diabetes was induced by a single intraperitoneal injection of 50 mg/kg streptozotocin (STZ, Sigma Aldrich) in citrate buffer (pH = 4.5) for consecutive 5 days. One week after STZ injection, the blood glucose levels of mice were continuously measured for 7 days, and those exhibiting levels >16.7 mM were considered diabetic[56].

Subsequently, full-layer skin wounds were established in all diabetic mice (approximately 10 mm in diameter), which were coated with a bacterial mixture (100 µL) of SA and PA ($10^8$ CFU/mL). The chronic mouse model of diabetic wounds was considered to be successfully established when biofilms in wounds were formed after 2 days. The mice were randomly divided into the following four groups: control group, C@PA MN(+) group, Se@PA MN group, and SeC@PA MN(+) group. After 4, 8, 12, and 16 days, digital images of wounds were acquired and the wound area was calculated using ImageJ. The percentage of wound size was calculated as follows: $[A_{(4,8,12,16)}/A_0] \times 100\%$, where $A_0$ and $A_{(4,8,12,16)}$ represent the wound areas on day 0 and days 4, 8, 12, and 16, respectively.

The pathological status and healing process of wounds were evaluated by H&E staining and Masson's trichrome staining. Bacteria were characterized using Gram staining and counted using plate-counting assays. RS and CD31 expression were assessed using immunohistochemical staining and F4/80, CD86, and CD206 levels were determined using immunofluorescence staining.

## Statistical analysis

The data here were presented as mean ± standard deviation. The statistical analyses were performed using GraphPad Prism 7.0 (La Jolla, CA, USA) software. Differences between two groups were analyzed by Student's $t$ test. Comparisons among the groups were performed using one-way ANOVA with Tukey's multiple comparison test. In all cases, statistical significance was considered when $p < 0.05$.

## Reporting summary

Further information on research design is available in the Nature Portfolio Reporting Summary linked to this article.

## Data availability

All other data are available from the corresponding authors upon request. Source data are provided with this paper.

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

## Acknowledgements

We are grateful for the financial support from the National Natural Science Foundation of China (32071342 and 32101065), the Science, Technology & Innovation Commission of Shenzhen Municipality (JCYJ20220818102810023), and the Singapore National Research Foundation under Its Competitive Research Program (NRF-CRP26-2021-0002).

## Author contributions

X.W.Z, Y.L.Z and H.Z.C supervised the project. L.Y. designed the study and conceived the paper in discussions with X.W.Z., Y.L.Z. and H.Z.C. D.Z., W.J.L., C.D.D., L.L.W., Q.Y.L., H.B.L., Z.M.L. and L.M. contributed to the laboratory assistance. All authors read and approved the final manuscript.

## Competing interests

The authors declare no competing interests.
