## [Peer Review File · Nature Communications]

Reviewers' Comments:

Reviewer #1:

Remarks to the Author:

In this manuscript, Yang et al. developed a promising microneedle (MN) bandage for treating diabetic wounds with biofilm infections. This MN therapeutic platform contains dopamine-coated nanohybrid with selenium, Ce6, and L-arginine (LA), which possess the abilities of physical penetration of drugs and chemical regulation of the ROS level. MN bandage can effectively deliver the loadings into the interior of biofilm-infected tissues and reduce the GSH by Se and NO. Both in vitro and in vivo experiments demonstrated the antibiofilm effect on SA and PA. The design strategy is interesting by using selenium to regulate the ROS generation/elimination according to the microenvironment, which may inspire the design of PDT agents with high efficiency and low side effects for treating bacterial biofilms. However, some issues should be addressed to improve this manuscript. The concerns are as follows.

1. In this study, many ROS and RNS are involved and may contribute to antibacterial effects. For instance, Se can mediate the hydroxyl generation and effectively kill the bacteria. While NO may also form RNS, as indicated in Figure 1, which potentially has antibacterial activity, but was not thoroughly studied. Moreover, the contribution of $1O_2$, $O_2^{\bullet-}$, which were proven antibacterial species in previous reports, has not been evaluated in this study. Hence, the exact antibacterial mechanism for this study seems blurry. It is suggested to make it clearer through further experiments, such as ROS quenching assay.
2. The loading efficiency or encapsulating efficiency of Se, Ce6, and LA in SeC@PA should be evaluated. Also, what about the stability of SeC@PA? As mentioned, the LA was adsorbed on the surface, which may leak during storage.
3. As the basis of this study, the GSH level in different parts of the infected tissues is a crucial factor for the ROS regulation of SeC@PA. However, no detailed discussion or direct experimental data show the GSH level in the infected tissues, which may significantly weaken the conclusion. So, it is suggested to investigate the exact GSH concentration in tissues to more accurately describe the "High-GSH" or "Low-GSH".
4. For the material preparation, Se nanoparticles were coated with a thick layer of PDA. As mentioned, the Se and Ce6 can be released from the SeC@PA. Nevertheless, the degradation of PDA was not discussed, and the release of Se was not proved by experiments. Previous reports showed that PDA could remove ROS. Is that possible in this study?
5. LA was used to generate NO, which can further deplete GSH and generate RNS. What about the direct antibiofilm effect of NO? If the RNS was produced, as illustrated in Figure 1, what is the role played by it for biofilm eradication? Since many works have demonstrated the biofilm elimination effect of NO, the authors should compare the effect of NO in this study to the other works.

Reviewer #2:

Remarks to the Author:

The present article reported a MN bandage functionalized with dopamine-coated hybrid nanoparticles containing selenium and chlorin e6. As indeed stated by the authors, such bandage was capable of the dual-directional regulation of ROS generation by utilizing the wound microenvironment. However, the materials and device structure lack innovation to some extent, some of the data in the article are unconvincing and should be further explained. Furthermore, there are also some details not clear and should have been corrected/clarified.

Several issues are addressed as follows:

The used materials, dopamine-coated hybrid nanoparticles containing selenium and chlorin e6 (SeC@PA), are common and simple, moreover, a lot of relevant work has been published. Therefore, the novelty of this work is doubtful.

The author highlighted the bidirectional ROS regulation by SeC@PA, which means produce ROS at high GSH level and scavenge ROS at low GSH level. But only conditions in 8 mM and 2 μ M of GSH

were exploited, what is the turning point or plateau for change of ROS regulation? if it is turning point, how to guarantee the biofilm is totally eradicated and the subsequent lower ROS will not weaken the anti-biofilm property, if it is plateau, the stated bidirectional ROS regulation can not be achieved.

I think there is a problem with English expression for example: "an SEM accessory" (line 5, page 5), "an ROS storm" (line 9, page 5). The manuscript must be reviewed by a native person with expertise in the review of scientific texts.

Figure 2F, peaks of Se⁴⁺ (58.20 eV and 59.06 eV) are both 3d^{5/2}, peaks for Se⁰ (54.73 eV and 55.53 eV) are 3d^{5/2} and 3d^{3/2}? Please reconfirm this point.

Figure 3A just presented the image of live/dead stain of SA biofilms, detailed analysis of related parameters such as live dead/live bacteria ratio and thickness of is necessary to confirm the anti-biofilm property.

"The CFU values in the SeC@PA(+) group were reduced by 4, 3, and 1 orders of magnitude, respectively, compared with that in other groups."(line 1,page 9) this sentence doesn't express clearly.

The anti-biofilm property of PA is lack of in-depth discussion. As shown in Figure 3 and S10, the anti-biofilm property of PA is weaker than that of SA, such phenomenon should be explained.

Figure 4, it is suggested that the author adjust the order of figure consistent with the discussion for reader to understand better.

Part for Antioxidant Properties of SeC@PA, what is the level of GSH in this part of experiment, how to maintain such level of GSH?

The migration test, why the induced HUVEC can not secret VEGF? Moreover, it is not clear that the HUVEC migrated rather than RAW264.7 in the Figure 5F .

Figure 6A,the initial wound areas for different groups seem not the same,this can be a problem.

Figure 7B showed lower ROS and Figure 7D presented fewer colonies for the SeC@PA MN (+) group, this is not consistent with discussion of Part Anti-biofilm property of SeC@PA, which concluded that the SeC@PA(+) group could eradicate biofilms by decomposing GSH to generate higher levels of ROS.

Reviewer #3:

Remarks to the Author:

This is an interesting manuscript that describes the fabrication and characterization of a microneedle bandage carrying dopamine coated hybrid nanoparticles containing selenium and chlorin e6 (SeC@PA). The material in presence of light generates or absorbs ROS depending on the environmental condition. Therefore, it could act as both antibacterial and anti-inflammatory. The properties of the MNs are accessed in vitro and later in a rodent model of type 1 diabetes with infected wounds. The level of characterization is excellent and results are properly justified. However, there are several important concerns that should be clearly addressed to assess if it is suitable for publication in Nat. Comm.

- There is a concern with lack of innovation. There are several publications on the use of phototriggered anti-bacterial and anti-inflammatory MNs. For example, RSC Adv., 2023, 13, 9998-10004 is describing a similar composition. Authors should clearly describe the difference between the present study and other publications.
- The claims that the material is suitable for less inflamed and non-infected animals are not tested. Proper animal studies should be performed to investigate the benefit of the MNs in such animal models.
- The in vitro results and the animal studies show very limited difference between different groups. It is not clear why authors believe the final design is more successful.
- The benefit of MNs are not demonstrated. Also, the MNs are small and typically cannot penetrate biofilms in the depth of tissue in humans. Authors should describe how this potentially can be used in real wounds. Also, the benefit of MNs over topical treatment should be demonstrated. Also, the effect of photo illumination alone should be tested. A group studying the effect of materials

without photo-illumination should also be studied.

- Figure captions should be expanded and stand alone. The groups should be described in the captions.

- Most chronic wounds are basic. Authors should also perform the in vitro studies in a basic environment.

Point-by-Point Response to Reviewers

To Reviewer #1 (Remarks to the Author):

In this manuscript, Yang et al. developed a promising microneedle (MN) bandage for treating diabetic wounds with biofilm infections. This MN therapeutic platform contains dopamine-coated nanohybrid with selenium, Ce6, and L-arginine (LA), which possess the abilities of physical penetration of drugs and chemical regulation of the ROS level. MN bandage can effectively deliver the loadings into the interior of biofilm-infected tissues and reduce the GSH by Se and NO. Both in vitro and in vivo experiments demonstrated the antibiofilm effect on SA and PA. The design strategy is interesting by using selenium to regulate the ROS generation/elimination according to the microenvironment, which may inspire the design of PDT agents with high efficiency and low side effects for treating bacterial biofilms. However, some issues should be addressed to improve this manuscript. The concerns are as follows.

Response: We express our sincere thanks to the reviewer for the positive comments and valuable suggestions, which greatly help us further improve the quality of this work.

Question 1: In this study, many ROS and RNS are involved and may contribute to antibacterial effects. For instance, Se can mediate the hydroxyl generation and effectively kill the bacteria. While NO may also form RNS, as indicated in Figure 1, which potentially has antibacterial activity, but was not thoroughly studied. Moreover, the contribution of $^1\text{O}_2$, $\text{O}_2^{\bullet-}$, which were proven antibacterial species in previous reports, has not been evaluated in this study. Hence, the exact antibacterial mechanism for this study seems blurry. It is suggested to make it clearer through further experiments, such as ROS quenching assay.

Response: Thank you for your valuable comments. We would like to apologize for any confusion arising from the collective use of "ROS" to refer to ROS and RNS in the original manuscript. We acknowledge this mistake and have rectified it in the revised manuscript by replacing "ROS" with reactive species (RS).

Three major RS were produced by SeC@PA(+): singlet oxygen ($^1\text{O}_2$) produced by Ce6, active nitrogen species (RNS) formed by NO, and hydroxyl radicals ($\bullet\text{OH}$) produced by Se. However, only in the presence of $^1\text{O}_2$ can LA coating on the outside of the SeC@PA produce NO, resulting in the subsequent generation of RNS. Se generated $\bullet\text{OH}$ through a cascade reaction in the presence of $^1\text{O}_2$. Therefore, we cannot use RS quenching approach to exclusively generate RNS or $\bullet\text{OH}$ within the system. Consequently, we employed the RS superposition method instead of RS quenching to investigate the individual contribution of each type of RS in combating the biofilms. Specifically, to simulate high GSH levels in biofilms, we added 8 mM GSH to each group:

(1) Control + 8 mM GSH group: No RS was generated after irradiation;

- (2) C@P + 8 mM GSH group: only $^1\text{O}_2$ was generated after irradiation;
- (3) C@PA + 8 mM GSH group: $^1\text{O}_2$ and RNS were generated after irradiation;
- (4) SeC@PA + 8 mM GSH group: $^1\text{O}_2$, RNS and $\bullet\text{OH}$ were generated after irradiation.

Considering the scavenging effect of GSH on RS, we adjusted the concentration of nanoparticles in each group to achieve comparable $^1\text{O}_2$ and RNS intensity (Please refer to Supplementary Information, Page S2 for detailed methods). As shown in Supplementary Fig. 13a, $^1\text{O}_2$ was generated in C@P + 8 mM GSH group, C@PA + 8 mM GSH group and SeC@PA + 8 mM GSH group, RNS was produced in C@PA + 8 mM GSH group and SeC@PA + 8 mM GSH group, while $\bullet\text{OH}$ was only generated in SeC@PA + 8 mM GSH group, and the intensities of $^1\text{O}_2$ and RNS exhibited no significant differences (Supplementary Fig. 11), thereby establishing a solid basis for the superposition technique. Subsequently, we explored the anti-biofilm efficacy of different groups. As presented in Supplementary Fig. 13b,d, the order of DCF fluorescence intensity in SA biofilm among different groups was SeC@PA(+) group > C@PA(+) group > C@P(+) group. The enhanced fluorescence intensity observed in the SeC@PA(+) group can be attributed to the combined effects of RNS and $\bullet\text{OH}$, which were superimposed compared to the C@P(+) group. Additionally, the SeC@PA(+) group exhibited a superposition of $\bullet\text{OH}$ compared to the C@PA(+) group. Supplementary Fig. 13c,e-f displayed live/dead biofilm images, crystal violet staining and bactericidal results, respectively. The results showed that the C@P(+) group exhibited a relatively intact biofilm and thus the anti-biofilm performance of this group was weak. In contrast, the C@PA(+) group showed the enhanced anti-biofilm effect with compromised biofilm integrity, which should be attributed to superposition of RNS, resulting in more robust anti-biofilm effect. Among all the groups, SeC@PA(+) group showed the strongest anti-biofilm effect, which was benefited from the superposition of RNS and $\bullet\text{OH}$. These results were consistent with the RS intensities in the biofilm.

Thus, this study investigated the involvement of three RS produced in SeC@PA in the anti-biofilm effect, which was termed as "RS storm". The anti-biofilm contributions of the three RS were quantified by crystal violet staining experiments, which represent the biomass of the biofilm. As shown in the Supplementary Fig. 14, the contribution of $^1\text{O}_2$, RNS and $\bullet\text{OH}$ against biofilm was 18.5%, 37.6% and 30.2%, respectively.

Supplementary Fig. 12. a Fluorescence intensity of SOSG. **b** O53 in different group (n = 4, mean \pm SD).

Supplementary Fig. 13. Anti-biofilm effect of different RS. **a** Fluorescence intensity of SOSG ($^1\text{O}_2$), O53 (RNS) and HPF ($\bullet\text{OH}$) in different group. **b** Detection of RS in the SA biofilm incubated with different nanoparticles after 660 nm irradiation (200 mW/cm²) for 3 min. Scale bar is 500 μm . **c** Live/dead stain of SA biofilms with different treatments under 660 nm irradiation (200 mW/cm²) for 3 min. **d** Semiquantitative statistics of the mean fluorescence intensity of DCF in SA biofilms treated with different groups (n=3, mean \pm SD). **P<0.01 compared with the indicated group, ***P<0.001 compared with the indicated group. **e** Representative images of crystal violet staining and absorbance values at 590 nm after treatment with different nanoparticles (n = 5, mean \pm SD). **P<0.01 compared with the indicated group, ***P<0.001 compared with the indicated group. **f** Bactericidal results of different nanoparticles characterized by the standard plate counting assay (n=3, mean \pm SD). *P<0.05 compared with the indicated group. C@P(+): Ce6-PDA nanoparticles under 660 nm irradiation (200 mW/cm²) for 3 min; C@PA(+): Ce6-PDA-LA nanoparticles under 660 nm irradiation (200 mW/cm²) for 3 min; SeC@PA(+): Se-Ce6-PDA-LA nanoparticles under 660 nm irradiation (200 mW/cm²) for 3 min.

Supplementary Fig. 14. Anti-biofilm contribution of $^1\text{O}_2$, RNS and $\bullet\text{OH}$.

To better express this point, we have modified the manuscript, as followed:

We supplemented fluorescence intensity of SOSG and O53 in different group as **Supplementary Fig. 12.**

We supplemented the anti-biofilm effect of different RS as **Supplementary Fig. 13.**

We supplemented the anti-biofilm contribution of different RS as **Supplementary Fig. 14.**

In Page 11, we added the statement: “To elucidate the “RS storm” strategy,”

Question 2: The loading efficiency or encapsulating efficiency of Se, Ce6, and LA in SeC@PA should be evaluated. Also, what about the stability of SeC@PA? As mentioned, the LA was adsorbed on the surface, which may leak during storage.

Response: Thank you for your questions and suggestions. We have supplemented experimental data and explanations to address your concerns.

(1) For the first question “The loading efficiency or encapsulating efficiency of Se, Ce6, and LA in SeC@PA should be evaluated.”

The loading efficiency of Se, Ce6, and LA in SeC@PA was provided in Supplementary Table 2.

(2) For the second question “Also, what about the stability of SeC@PA? As mentioned, the LA was adsorbed on the surface, which may leak during storage.”

Firstly, we investigated the particle stability of SeC@PA at two different pH values. The results were shown in Figure R1. The particle size of SeC@PA was relatively stable at pH=7.4, while significantly increased at pH=5.5. This difference was attributed to the degradation of the PDA layer in acidic condition.

Considering that the SeC@PA was loaded in the microneedle and stored in solid which could be preserved for a relatively long period, we examined the particle size of SeC@PA in microneedle on day 30. As shown in Figure R2, the SeC@PA loaded in the microneedle exhibited a slight increase in particle size after 30 days, reaching approximately 120 nm, which fell within an acceptable range.

Simultaneously, we assessed the leakage of LA from SeC@PA at different time intervals under pH=7.4 conditions. The results are presented in Figure R3. Over the initial nine days of storage, the leakage of LA gradually increased with time, eventually reaching a plateau. The leakage rate of LA remained relatively stable, remaining below 10% after one month of storage.

Figure R1. Size of SeC@PA at different time under different pH values.

Figure R2. Size of SeC@PA from SeC@PA MN at day 0 and day 30.

Figure R3. Percentage of LA released from SeC@PA at different times.

Supplementary Table 2. Loading efficiency of Se, Ce6 and LA in SeC@PA.

Se	Ce6	LA
23.23 ± 4.11 (%)	8.40 ± 4.38 (%)	12.60 ± 3.79 (%)

To better express this point, we have modified the manuscript, as followed:

In page 5, we added the statement: “The loading efficiency of Se, Ce6 and LA was 23.23 ± 4.11%, 8.40 ± 4.38% and 12.60 ± 3.79%, respectively (Supplementary Table 2).”

We supplemented the loading efficiency of Se, Ce6 and LA in SeC@PA as **Supplementary Table 2**.

Question 3: As the basis of this study, the GSH level in different parts of the infected issues is a crucial factor for the ROS regulation of SeC@PA. However, no detailed discussion or direct

experimental data show the GSH level in the infected tissues, which may significantly weaken the conclusion. So, it is suggested to investigate the exact GSH concentration in tissues to more accurately describe the "High-GSH" or "Low-GSH".

Response: Thank you very much for your valuable suggestions. The wound coverings containing both the biofilm and inflammatory exudates were removed from the model mice two days after biofilm infection. As a control, tissue beneath the wound coverings was extracted. GSH content in wound coverings and tissue were determined separately. As displayed in Supplementary Fig. 26, the GSH content in the wound coverings was much higher than that in the wound tissue, with a content as high as 6.69 mmol/g, which was 9.3 times higher than 0.72 mmol/g in the wound tissue.

To verify whether the bidirectional regulation of RS by SeC@PA could be achieved by the difference GSH level between wound tissue and wound coverings, we first examined the RS level in wound tissue and antimicrobial effects in wound coverings after 4 days treatment. As depicted in Fig. 9b,c, the SeC@PA MN(+) group exhibited significant clearance of reactive species (RS) in wound tissue when compared to the control group and C@PA MN(+) group. Additionally, the SeC@PA MN(+) group exhibited stronger antibacterial activity compared to the C@PA MN(+) group in wound coverings. These findings suggested that the GSH level in the wound tissue was low, and SeC@PA MN(+) played a role in clearing RS and alleviating oxidative stress in the wound tissue. Conversely, the GSH level in the wound coverings was high, and SeC@PA MN(+) could enhance RS to achieve a more potent anti-biofilm ability.

To further verify that this result was due to bidirectional regulation of RS by SeC@PA, rather than unidirectional regulation that increases with increasing GSH concentration, a range of GSH concentrations was established *in vitro* to assess RS production. As shown in Fig. 2e, SeC@PA(+) efficiently scavenged RS at low GSH concentrations (≤ 0.4 mM), while enhancing RS production at high concentrations (≥ 0.8 mM). A turning point was clearly observed, which indicated that SeC@PA successfully achieved bidirectional regulation of RS.

Supplementary Fig. 26. GSH content of tissue and wound coverings (n=5, mean \pm SD). ****P<0.0001 compared with the indicated group.

Fig. 9 b Representative immunofluorescence images of RS staining at day 4. Scale bar is 100 μ m. **c** Representative photographs of bacterial culture from the skin tissue of diabetic mouse wounds infected with biofilms at day 4.

Fig. 2e Effect of different GSH concentrations on the RS produced by SeC@PA(+) and C@PA(+) (n=4, mean ± SD).

To better express this point, we have modified the manuscript, as followed:

We supplemented the GSH content of tissue and wound coverings as **Supplementary Fig. 26**.

Fig. 2e in original manuscript was removed to Supplementary Information as **Supplementary Fig. 7**, and Effect of different GSH concentrations on the RS produced by SeC@PA(+) and C@PA(+) was added as **new Fig. 2e**.

In Page 21, we added the statement “Firstly, we determined GSH content in wound coverings (including biofilm and inflammatory exudates) and wound tissue (beneath the wound coverings)”

Question 4: For the material preparation, Se nanoparticles were coated with a thick layer of PDA. As mentioned, the Se and Ce6 can be released from the SeC@PA. Nevertheless, the degradation of PDA was not discussed, and the release of Se was not proved by experiments. Previous reports showed that PDA could remove ROS. Is that possible in this study?

Response: Thank you for pointing out these issues.

(1) For the first question “Nevertheless, the degradation of PDA was not discussed.”

We investigated the degradation of SeC@PA under different pH conditions. As shown in Figure R4, after being stored at pH=5.5 for 48 h, the particle size of SeC@PA changed obviously. However, no significant variation in particle size was observed at pH=7.4. This observation was further substantiated by the TEM images shown in Figure R5, which indicated a noticeable degradation of SeC@PA after 24 and 48 h at pH=5.5. This result aligned with our *in vitro* release results, confirming the consistency of the degradation trend.

(2) For second question “the release of Se was not proved by experiments.”

We investigated the release of Se from SeC@PA using Inductively Coupled Plasma-Atomic Emission Spectrometry (ICP-AES). The results were shown in Supplementary Fig. 5. At pH=5.5, the release of Se continuously increased with time, reaching a cumulative release of 56.39% at 48 h. Considering the slow release of Se from SeC@PA at pH=7.4 and the drug concentration might be lower than the detection limit, we only measured the Se concentration at 48 h. The cumulative release of Se at pH=7.4 for 48 h was 7.67%, significantly lower than the release observed at pH=5.5.

(3) For the third question “Previous reports showed that PDA could remove ROS. Is that possible in this study?”

To explore whether the PDA in this study has a scavenging effect on the RS, we performed a contrast experiment with C@PA using a physical mixture of Ce6 and LA. When compared to the physical mixture of Ce6 and LA, C@PA increased the PDA, which was in line with the principle of controlling a single variable. As shown in Figure R6, there was no significant difference in RS between the two groups. This indicated that PDA did not have a significant effect on clearing RS in this study.

Figure R4. Size distribution of SeC@PA under different conditions.

Figure R5. TEM images of SeC@PA under different conditions. The scale bar is 200 nm.

Supplementary Fig. 5 Cumulative release of Se from SeC@PA in pH=5.5 (n=3, mean \pm SD).

Figure R6. Mean fluorescence intensity of DCF in different group (n=4, mean \pm SD).

To better express this point, we have modified the manuscript, as followed:

We supplemented Cumulative release of Se from SeC@PA in pH=5.5 as **Supplementary Fig. 5**.

In Page 5, we added the statement “Therefore, we studied the release of Ce6 and Se from SeC@PA under different pH conditions. The release of Ce6 and Se from SeC@PA was accelerated in a weakly acidic environment (pH=5.5) (Supplementary Fig. 4 and Supplementary Fig. 5).”

Question 5: LA was used to generate NO, which can further deplete GSH and generate RNS. What about the direct antibiofilm effect of NO? If the RNS was produced, as illustrated in Figure 1, what is the role played by it for biofilm eradication? Since many works have demonstrated the biofilm elimination effect of NO, the authors should compare the effect of NO in this study to the other works.

Response: Thanks a lot for your questions.

(1) For the first question “What about the direct antibiofilm effect of NO”

C@PA was chosen for investigation because it can generate NO similar to SeC@PA without interfering with the reaction between NO and GSH. This provides a foundation for studying the role of NO in subsequent research systems. In SeC@PA, the presence of Se can react with GSH, leading to potential interference when studying the reaction between NO and GSH.

Initially, a high concentration (20 mM) of the reducing substance vitamin C (Vc) was used to eliminate ROS and RNS generated by C@PA(+). Figure R7 demonstrated that ROS and RNS in C@PA(+) were effectively eliminated after the addition of Vc, achieving the intended purpose. Subsequently, the anti-biofilm effect of C@PA(+) + 20 mM Vc was examined. As depicted in Figure R8A,B, the anti-biofilm effect of C@PA(+) + 20 mM Vc did not show a

significant difference compared to the control group. Based on this observation, we hypothesize that once NO is produced in this study, it reacts with GSH in the biofilm to form GSSG and with ROS to form RNS, without exerting its direct anti-biofilm effect.

In order to verify our hypothesis, we measured the production of RNS and GSSG of C@PA in the presence of 8 mM GSH at different irradiation time. As shown in Figure R8C,D, GSSG and RNS levels increased with irradiation time, which is in consistent with the NO production (Fig. 1b). In summary, the NO produced in this study mainly plays two roles: Firstly, it depletes GSH and amplifies the anti-biofilm effect of RS; Secondly, it produces more active RNS, enhancing the anti-biofilm effect. These findings are consistent with our initial expectation.

(2) For the second question “If the RNS was produced, as illustrated in Figure 1, what is the role played by it for biofilm eradication?”

The role of RNS played for biofilm eradication was studied. As displayed in Supplementary Fig. 14, $^1\text{O}_2$, RNS and $\bullet\text{OH}$ produced in this study actively participated in biofilm eradication. The respective proportions of $^1\text{O}_2$, RNS and $\bullet\text{OH}$ against biofilm in mitigating biofilm were 18.5%, 37.6% and 30.2%, respectively.

(3) For the third question “Since many works have demonstrated the biofilm elimination effect of NO, the authors should compare the effect of NO in this study to the other works”.

Indeed, the potent antimicrobial activity of NO is mainly dependent on the formation of RNS, such as nitrous oxide (N_2O_3) and peroxynitrite (ONOO^-). The RNS contributes to process such as lipid peroxidation, bacterial cell membrane disruption and DNA deamination. This has been mentioned in several studies (*Adv. Funct. Mater.* 2022, 32, 211148; *Adv. Healthcare Mater.* 2018, 7, e1800155; *Adv. Sci.* 2023, 10, e2206959). Therefore, the NO produced in this study mainly relies on the production of RNS to exert its anti-biofilm effect, which was consistent with the description in the previous studies.

Figure R7. Fluorescence intensity of DCF in different group.

Figure R8. Anti-biofilm effect of NO. (A) Representative images of crystal violet staining and absorbance values at 590 nm after treatment with different nanoparticles (n=5, mean ± SD). ***P<0.001 compared with the indicated group, ns no significant compared with the indicated group. (B) Bactericidal results of different nanoparticles characterized by the standard plate counting assay (n=3, mean ± SD). ***P<0.001 compared with the indicated group, ns no significant compared with the indicated group. (C) GSSG concentration of C@PA(+) at different irradiation times in the presence of 8 mM GSH. (D) RNS mean fluorescence intensity of C@PA(+) at different irradiation times in the presence of 8 mM GSH.

Fig. 1b Concentration of NO produced by SeC@PA with different irradiation intensity at 660 nm (n=3, mean ± SD).

Supplementary Fig. 14. Anti-biofilm contribution of different RS.

To Reviewer #2 (Remarks to the Author):

The present article reported a MN bandage functionalized with dopamine-coated hybrid nanoparticles containing selenium and chlorin e6. As indeed stated by the authors, such bandage was capable of the dual-directional regulation of ROS generation by utilizing the wound microenvironment. However, the materials and device structure lack innovation to some extent, some of the data in the article are unconvincing and should be further explained. Furthermore, there are also some details not clear and should have been corrected/clarified.

Response: Thanks for your useful comments and suggestions. We would like to apologize for any confusion caused by referring to ROS and RNS collectively as "ROS" in the original manuscript. We were aware of this mistake and have replaced "ROS" with reactive species (RS).

Several issues are addressed as follows:

Question 1: The used materials, dopamine-coated hybrid nanoparticles containing selenium and chlorin e6 (SeC@PA), are common and simple, moreover, a lot of relevant work has been published. Therefore, the novelty of this work is doubtful.

Response: We acknowledge the reviewer's concern regarding the novelty of the SeC@PA system described in this work. Indeed, polydopamine-based hybrid nanoparticles have been reported in a lot of work. However, it is worth noting that the highlights of this work lie in addressing the bottleneck of chronic wound treatment. Specifically, this study aims to address the neglected differences between biofilm and wound tissue microenvironment and the barrier effect of wound coverings, through a rational biological design and optimized delivery strategy. The highlights mainly focus on the following three aspects:

(1) Self-enhancing the anti-biofilm effect of PDT by exploiting the high GSH levels of biofilms. As mentioned on Page 2 "Photodynamic therapy (PDT), which is a less invasive procedure and not associated with drug resistance, has attracted extensive research in antibacterial therapy. However, there are challenges associated with the use of PDT to effectively treat biofilm infections, as the ROS generated by photosensitizers (PS) can be depleted by the high levels of GSH in biofilms. To address these issues, most studies focus on the depletion of endogenous GSH by mediators such as nitric oxide (NO)." However, the strategy to exploit high GSH levels in biofilms to enhance PDT has barely been developed. In this study, we utilized the high GSH levels present in biofilms, which react with SeC@PA to generate hydroxyl radicals ($\bullet\text{OH}$). Additionally, SeC@PA has the ability to deplete GSH and produce RNS. This self-enhancing strategy of generating more reactive species (RS) and decreasing GSH doubly amplified the anti-biofilm effect of PDT, which provides a novel biofilm microenvironment triggered self-enhancing strategy for the treatment of biofilm infections.

(2) Bidirectional regulation of RS exploiting the different GSH levels between biofilms and wound tissue is a crucial aspect that we highlight in our manuscript (Page 3): "Another point of concern is that ROS is a double-edged sword. Wound beds covered by biofilms are

excessively inflammatory, and oxidative stress in wounds can be aggravated by the ROS produced during PDT used to destroy biofilms, leading to impaired wound healing. Therefore, the prompt elimination of ROS during PDT to prevent further aggravation of wound inflammation, and increasing the antioxidative capacity of the wound beds to endow upon wounds an anti-inflammatory effect from the outside to the inside are of utmost importance.” In our work, we employed SeC@PA to exploit the different GSH levels in wound tissue and biofilms enabling bidirectional regulation of RS (including ROS and RNS). In the wound tissue with low GSH levels, SeC@PA functions to clear RS and enhances the content and activity of GPX4, thereby exhibiting a strong antioxidant capacity.

To further support the significance of bidirectional RS regulation, we conducted additional experiments. Firstly, we measured GSH concentrations in wound tissue and wound coverages, including biofilms and inflammatory exudates. The results, presented in Supplementary Fig. 26, demonstrated a significant difference in GSH levels between the two sites. The GSH content in the wound coverings was significantly higher than that in the wound tissue, with a content as high as 6.69 mmol/g, which was 9.3 times higher than 0.72 mmol/g in the tissue. These results make it possible for SeC@PA to bidirectionally regulate RS between biofilms and wound tissue.

Secondly, we employed a diabetic wound model without biofilms infection to verify the direct damaging effect of PDT on wound tissue. As shown in Figure R9, the C@PA (+) group seriously hindered the wound healing, attributed to the more serious oxidative stress at the wound. On the contrary, the SeC@PA (+) group significantly promoted wound healing and significantly reduced oxidative stress in wound tissue. These findings highlight the potential side effects of RS-induced oxidative stress during PDT treatment for chronic wounds with biofilm infection. In our work, this issue was addressed by the bidirectional regulatory effect of SeC@PA on RS. This bidirectional regulation strategy provides a new way for the wound treatment of biofilm infection.

(3) Optimized delivery strategy. As we stated in original manuscript (Page 3): “Moreover, an important factor affecting the efficacy of therapeutics is their point of delivery. The importance of the role of the delivery point in promoting wound healing has been well studied. Chronic wounds are typically covered with eschar and biofilms, which constitute a physiochemical barrier hindering the penetration of topical therapeutic agents into the underlying live tissue. Thus, when drugs are used topically, their local bioavailability is reduced, as expected. A delivery system that can disrupt the physiochemical barriers created by eschar and biofilms and shorten the travel distance of therapeutics by directly delivering them to live tissue can improve drug efficacy.” In our work, microneedles facilitate to break the barrier formed by biofilms and inflammatory exudates, contributing to the effective delivery of SeC@PA. We compared the therapeutic effects of SeC@PA MN and SeC@PA NPs, as well as their penetration efficiency into wound coverings (Fig. 7), to underscore the significance of microneedles in the treatment of chronic wounds with biofilm infection.

Overall, this work highlights the self-enhancement strategy triggered by biofilm microenvironment, bidirectional RS regulation strategy, and optimized delivery strategy to address the current bottlenecks in PDT treatment of biofilm-infected chronic wounds.

Supplementary Fig. 26 GSH content of tissue and wound coverings (n=5, mean \pm SD). ****P<0.0001 compared with the indicated group.

Figure R9. Photodynamic therapy alone delays wound healing. (A, C) Representative images of wounds during healing and quantitative data of relative wound area to day 9 of the different groups at different time points (n=6, mean ± SD). Scale bar is 5 mm. *P<0.05 compared with the indicated group, #P<0.05 compared with other groups. **(B, D)** Representative immunofluorescence images of RS staining and relative fluorescence intensity of RS at day 2 (n=3, mean ± SD). Scale bar is 100 μm. *P<0.05 compared with the other groups.

Fig. 7 SeC@PA MN for promoting chronic wound healing through efficient delivery of SeC@PA NPs. **a, g** Representative images of wounds during healing and quantitative data of relative wound area to day 16 of the different groups at different time points ($n=4$, mean \pm SD). Scale bar is 5 mm. $*P<0.05$ compared with the indicated group. **b** Representative H&E staining images of wound samples treated with different groups on day 16. Scale bar is 100 μm . **c, h** Representative Masson's trichrome-stained images of wound samples and quantification of the collagen volume fraction for four groups on day 16 ($n=3$, mean \pm SD). Scale bar is 500 μm . $**P<0.01$ compared with the indicated group. **d, i** Representative immunofluorescence images of CD31 staining in the regenerated skin tissue and quantification of blood vessel density on day 9 after wound healing ($n=3$, mean \pm SD). Scale bar is 100 μm . $**P<0.01$ compared with the indicated group. **e** Representative images of distribution of SeR@NPs in wound coverings at different time in SeR@PA MN group. Scale bar is 500 μm . **f** Representative images of distribution of SeR@NPs in wound coverings at different time in SeR@PA NPs group. Scale bar is 500 μm . **j** Schematic representation of the acquisition of ex vivo wound coverages and determination of penetration efficiency of SeR@PA in wound coverings. **k** Penetration efficiency of SeR@PA MN and SeR@PA NPs in wound coverings ($n=3$, mean \pm SD). SeC@PA NPs(+): Solution containing Se-Ce6-PDA-LA nanoparticles under 660 nm irradiation (200 mW/cm²) for 3 min; SeC@PA MN(+): Microneedle containing Se-Ce6-PDA-LA nanoparticles

under 660 nm irradiation (200 mW/cm²) for 3 min; SeR@PA NPs: Solution containing Se-Rhodamine-PDA-LA nanoparticles; SeR@PA MN: Microneedle containing Se-Rhodamine-PDA-LA nanoparticles.

To better exhibit our novelty, we have modified the manuscript, as followed:

We added treatment of non-biofilm infected wounds in diabetic mice using SeC@PA bandages as **Fig. 6**.

We added SeC@PA MN promoted chronic wound healing by efficient delivery of SeC@PA as **Fig. 7**.

In Page 18, we added the statement “Diabetic chronic wounds without infected biofilm were employed to verify the direct RS-scavenging....”

In Page 19, we added the statement “Chronic diabetic wounds are mostly covered by inflammatory exudates and biofilms,”

Question 2: The author highlighted the bidirectional ROS regulation by SeC@PA, which means produce ROS at high GSH level and scavenge ROS at low GSH level. But only conditions in 8 mM and 2 μ M of GSH were exploited, what is the turning point or plateau for change of ROS regulation? if it is turning point, how to guarantee the biofilm is totally eradicated and the subsequent lower ROS will not weaken the anti-biofilm property, if it is plateau, the stated bidirectional ROS regulation cannot be achieved.

Response: We appreciate your insightful question and acknowledge that we should have provided more data regarding the turning point or plateau for the change in ROS regulation in different GSH levels.

(1) For the first question “what is the turning point or plateau for change of ROS regulation?”

We explored the effect of different GSH concentrations on RS production by SeC@PA(+) and C@PA(+) to find the turning point or plateau of GSH concentration. As shown in Fig. 2e, the intensity of RS in the C@PA(+) group decreased with increasing GSH concentration, which was attributed to the scavenging effect of GSH on the RS. In contrast, SeC@PA(+) played a role in scavenging RS at low GSH concentrations (≤ 0.4 mM), while enhancing RS intensity at high concentrations (≥ 0.8 mM), a clear turning point was observed.

We speculated that the bidirectional regulation of SeC@PA(+) on RS could be attributed to the dual role of Se in the system. One is to clear RS, and the other is to produce more active RS such as \bullet OH through a cascade reaction with GSH. When the GSH concentration is low, the production of RS is limited and could be scavenged by Se and GSH in the system. Therefore, SeC@PA(+) primarily serves as a scavenger of RS. On the other hand, when the GSH concentration is high, the system produced an excess of RS which exceeded the scavenging capacity of the system, resulting in the effect of enhancing RS. However, as the GSH concentration increases, the scavenging capability of RS in the system also increased. Therefore, we observed that at high GSH concentrations, RS intensity did not always increase

with the increase of GSH concentration; instead, it started to decrease at 4 mM. Therefore, SeC@PA (+) exhibited a dynamic regulation of RS production at different GSH concentrations.

(2) For the second question “if it is turning point, how to guarantee the biofilm is totally eradicated and the subsequent lower ROS will not weaken the anti-biofilm property”

We investigated the variation of RS in different nanoparticles with different irradiation time to explain how SeC@PA achieved the potent anti-biofilm effect. As illustrated in Figure R10A, the RS levels generated by the SeC@PA(+) group were comparable to those of the C@PA(+)+GSH group, but significantly lower than that of C@PA(+) group, which again revealed the RS scavenging capacity of Se. Interestingly, the SeC@PA(+)+GSH group showed a RS-enhanced effect during the initial 210 s of irradiation. However, after 240 s, the RS intensity began to decrease and was lower than that of the C@PA(+), indicating that the SeC@PA(+)+GSH group played a role in scavenging RS after 240 s of irradiation. To further verify whether this situation was caused by a change in GSH concentration, we determined the GSH concentration of 0.88 mM and 0.68 mM (Figure R10B) in the SeC@PA(+)+GSH group after 210 s and 240 s irradiation, respectively, which indicated that GSH concentration ranged between 0.68 mM and 0.88 mM, SeC@PA(+) changed the regulation of RS. Based on the results in Fig. 2e, it can be concluded that 0.88 mM represented a high GSH concentration which can achieve the RS-enhancing effect.

It should be noted that SeC@PA(+)+GSH showed RS enhancement within irradiation durations of less than 210 s. In fact, the irradiation time of our experiments was 180 s (3 min). Therefore, it can be concluded that SeC@PA(+) played a role in enhancing RS production during irradiation without diminishing RS intensity compared with C@PA(+). Moreover, the decline rate of GSH is consistent with the data in the original manuscript (Fig. 2i). Overall, SeC@PA(+) showed a strong anti-biofilm effect by enhancing RS production through 3 min of irradiation at a high GSH concentration of the biofilm.

Fig. 2e Effect of different GSH concentrations on the RS produced by SeC@PA(+) and C@PA(+)

(n=4, mean \pm SD).

Figure 10. (A) DCF fluorescence intensity of different nanoparticles at different irradiation times (n=4, mean \pm SD). (B) Concentration of GSH in SeC@PA(+)+GSH group after 210 s and 240 s irradiation (n=3, mean \pm SD).

To better express this point, we have modified the manuscript, as followed:

The Fig. 2e in original manuscript was removed to Supplementary Information as **Supplementary Fig. 7**, and Effect of different GSH concentrations on the RS produced by SeC@PA(+) and C@PA(+) was added as **new Fig. 2e**.

In Page 7, we added the statement “Subsequently, we explored the impact of varying concentrations of GSH on the RS ...”

Question 3: I think there is a problem with English expression for example: “an SEM accessory” (line 5, page 5), “an ROS storm” (line 9, page 5). The manuscript must be reviewed by a native person with expertise in the review of scientific texts.

Response: Thanks for your advice. We have carefully revised our manuscript following your advice. For example, we have modified the “an SEM accessory” to “the SEM accessory”, “an ROS storm” to “RS Storm”. The English language of this paper was edited by LetPub, specializes in editorial services for the scholarly publishing community and provides research communications services (<https://www.letpub.com/>).

Question 4: Figure 2F, peaks of Se 4^+ (58.20 eV and 59.06 eV) are both 3d $_{5/2}$, peaks for Se 0 (54.73 eV and 55.53 eV) are 3d $_{5/2}$ and 3d $_{3/2}$? Please reconfirm this point.

Response: Thank you very much for your kind reminder. We sincerely apologize for the mislabeling of the aforementioned Se 4^+ peaks (59.06 eV) as 3d $_{5/2}$. We have corrected it as 3d $_{3/2}$. The Fig. 2f in original manuscript has also been corrected.

Question 5: Figure 3A just presented the image of live/dead stain of SA biofilms, detailed

analysis of related parameters such as live dead/live bacteria ratio and thickness of is necessary to confirm the anti-biofilm property.

Response: Thanks a lot for your advice. We have added the live dead/live biomass ratio and thickness of SA biofilm and PA biofilm as Supplementary Fig. 16 and Supplementary Fig. 17, respectively.

Supplementary Fig. 16 a Live/dead biomass ratio of SA biofilm with different treatments (n=3, mean ± SD). b Thickness of SA biofilm after different treatments (n=3, mean ± SD). ***P<0.001 and **P<0.01 compared with the indicated group.

Supplementary Fig. 17 a Live/dead biomass ratio of PA biofilm with different treatments (n=3, mean ± SD). b Thickness of PA biofilm after different treatments (n=3, mean ± SD). ***P<0.001 compared with the indicated group.

To better express this point, we have modified the manuscript, as followed:

We added the Live/dead biomass ratio and Thickness of SA biofilm as **Supplementary Fig. 16**.

We added the Live/dead biomass ratio and Thickness of PA biofilm as **Supplementary Fig. 17**.

Question 6: “The CFU values in the SeC@PA(+) group were reduced by 4, 3, and 1 orders of magnitude, respectively, compared with that in other groups.”(line 1,page 9) this sentence doesn’t express clearly.

Response: We apologize for any lack of clarity in expressing the results. We have revised this sentence as “The CFU values in the SeC@PA(+) group were reduced by 4, 3, and 1 orders of magnitude compared to control group, Se@PA group and C@PA(+) group, respectively.”

Question 7: The anti-biofilm property of PA is lack of in-depth discussion. As shown in Figure 3 and S10, the anti-biofilm property of PA is weaker than that of SA, such phenomenon should be explained.

Response: Thank you for your valuable comments. The original manuscript (Pages 8-10) extensively discussed the antimicrobial effects of SeC@PA (+) on SA and PA biofilms. The anti-biofilm effect of SeC@PA was confirmed by live/dead biofilm staining, crystal violet staining, and dilution plate assay. Among them, the dilution plate assay provided the most compelling evidence of antibacterial activity. We performed plate counts after SeC@PA(+) treatment of PA biofilms and SA biofilms, which were both ultimately reduced by 4 orders of magnitude relative to the control group. Therefore, based on our experimental results, we suggest that SeC@PA(+) has comparable anti-biofilm effects on PA biofilm and SA biofilm.

Question 8: Figure 4, it is suggested that the author adjust the order of figure consistent with the discussion for reader to understand better.

Response: Thanks a lot for your helpful suggestion. We are sorry for the inconvenience caused by the unreasonable layout of the Fig. 4. We have adjusted the order consistent with the discussion.

Fig. 4 Anti-inflammatory effect of nanoparticles. **a, b** Detection of RS by fluorescence of DCF in the HUVECs incubated with different nanoparticles under 660 nm irradiation (200 mW/cm²) for 3 min. Scale bar is 500 μm. ****P*<0.001 compared with the indicated group. **c** Intracellular DCF fluorescence intensity of HUVECs with different groups analyzed by flow cytometry. **d** Representative images of GPX4 staining of HUVECs after different treatments. Scale bar is 200 μm. **e, f** Expression level of GPX4 in the HUVECs determined by western blotting after treatment (*n*=3, mean ± SD). **P*<0.05 and ***P*<0.01 compared with the indicated group. **g** Activity of GPX4 treated with different groups (*n*=3, mean ± SD). ****P*<0.01 compared with the indicated group. Se@PA: Se-PDA-LA nanoparticles; C@PA(+): Ce6-PDA-LA nanoparticles under 660 nm irradiation (200 mW/cm²) for 3 min; SeC@PA(+): Se-Ce6-PDA-LA nanoparticles under 660 nm irradiation (200 mW/cm²) for 3 min.

Question 9: Part for Antioxidant Properties of SeC@PA, what is the level of GSH in this part of experiment, how to maintain such level of GSH?

Response: thanks a lot for your helpful questions.

(1) For the first question “Part for Antioxidant Properties of SeC@PA, what is the level of GSH in this part of experiment”.

The antioxidant properties of SeC@PA were assessed in HUVECs. We determined the GSH content in HUVECs was 0.405±0.008 mM/10⁹ cells. As displayed in Fig. 4, compared with the C@PA(+) group, the intensity of RS in SeC@PA(+) was much reduced, which suggested that SeC@PA acted as a scavenger of RS at this GSH level. Thus, GSH levels in HUVECs are what we claim to be low GSH level.

(2) For the second question “how to maintain such level of GSH”.

According to the results shown in Fig. 2i and Fig. 3g, SeC@PA has the ability to consume GSH under irradiation, which indicated that the treatment of SeC@PA(+) would lead to a decrease in GSH content. Therefore, GSH content in HUVECs would maintain low GSH level throughout the irradiation and SeC@PA played a role in scavenging RS.

Question 10: The migration test, why the induced HUVEC can not secrete VEGF? Moreover, it is not clear that the HUVEC migrated rather than RAW264.7 in the Figure 5F.

Response: Thank you for your helpful questions.

(1) For the first question “The migration test, why the induced HUVECs can not secrete VEGF?”.

There seems to be a misunderstanding. In the original manuscript (Page 16), we stated that “At the same time, there was no significant change in HUVECs,” which means that in HUVECs, the VEGF level in the SeC@PA(+) group was comparable to that in the control group, and no significant increase occurred, rather than that HUVECs did not secrete VEGF. For better understanding, we have revised the original manuscript as “On the contrary, the VEGF level in HUVECs treated with SeC@PA (+) group exhibited no obvious difference from that in the

control group."

(2) For the second question "Moreover, it is not clear that the HUVEC migrated rather than RAW264.7 in the Figure 5F."

We apologize for the ambiguous description of the experimental procedure. In fact, we used a transwell with a pore size of 0.4 μm to culture RAW264.7, which was then placed into a 6-well plate which culture with HUVECs (Supplementary Fig. 24). The pore size of transwell was smaller than the size of RAW264.7 cells, which ensured that RAW264.7 could not migrate into the well plate. The same procedure was used in the tube formation experiments. To express the experimental process more clearly, we have added experimental schematics to the methodology of the manuscript (Page 35).

Supplementary Fig. 24. a Schematic representation for migration assay of HUVECs. b Schematic diagram for the tube formation assay of HUVECs.

Question 11: Figure 6A, the initial wound areas for different groups seem not the same, this can be a problem.

Response: We appreciate your question. Our modeling protocol involved a two-day biofilm growth period. During this period, mice showed inconsistent wound contraction due to individual differences, resulting in different wound sizes. This phenomenon has also been observed in other studies (*Nat. Commun.* 2022, 13, 23875; *ACS Nano* 2019, 13, 11686). We provided initial wound area data for each group. As shown in Figure R11, there was no obvious difference in the mean wound area for each group. For Fig. 8a, the most representative wounds of each group were selected for presentation.

Figure R11. Initial wound area of different group (n=6. mean \pm SD).

Question 12: Figure 7B showed lower ROS and Figure 7D presented fewer colonies for the SeC@PA MN (+) group, this is not consistent with discussion of Part Anti-biofilm property of SeC@PA, which concluded that the SeC@PA(+) group could eradicate biofilms by decomposing GSH to generate higher levels of ROS.

Response: Thank you for your insightful comment. The RS depicted in Fig. 7b represented the RS found in the wound tissue (beneath the wound coverings, low GSH levels) rather than the wound coverings (including biofilms, high GSH levels). Therefore, the lower RS depicted in Fig. 7b and the fewer colonies in SeC@PA group in Fig. 7d can be attributed to the enhanced RS level in biofilm (high GSH levels) and decreased RS level in wound tissue (low GSH levels) of SeC@PA(+). This finding was consistent with the conclusion obtained in the SeC@PA anti-biofilm part that SeC@PA could eradicate biofilms (high GSH level) by decomposing GSH to generate higher levels of RS.

To Reviewer #3 (Remarks to the Author):

This is an interesting manuscript that describes the fabrication and characterization of a microneedle bandage carrying dopamine coated hybrid nanoparticles containing selenium and chlorin e6 (SeC@PA). The material in presence of light generates or absorbs ROS depending on the environmental condition. Therefore, it could act as both antibacterial and anti-inflammatory. The properties of the MNs are accessed in vitro and later in a rodent model of type 1 diabetes with infected wounds.

The level of characterization is excellent and results are properly justified. However, there are several important concerns that should be clearly addressed to assess if it is suitable for publication in Nat. Comm.

Response: We express our sincere thanks to the reviewer for the positive comments, and we have carefully revised manuscript according to the suggestions.

Question 1: There is a concern with lack of innovation. There are several publications on the use of phototriggered anti-bacterial and anti-inflammatory MNs. For example, RSC Adv., 2023, 13, 9998-10004 is describing a similar composition. Authors should clearly describe the difference between the present study and other publications.

Response: We acknowledge the reviewer's concern regarding the novelty of the SeC@PA system described in this work. Indeed, polydopamine-based hybrid nanoparticles have been reported in a lot of work. However, it is worth noting that the highlights of this work lie in addressing the bottleneck of chronic wound treatment. Specifically, this study aims to address the neglected differences between biofilm and wound tissue microenvironment and the barrier effect of wound coverings, through a rational biological design and optimized delivery strategy. The highlights mainly focus on the following three aspects:

(1) Self-enhancing the anti-biofilm effect of PDT by exploiting the high GSH levels of biofilms. As mentioned on Page 2 "Photodynamic therapy (PDT), which is a less invasive procedure and not associated with drug resistance, has attracted extensive research in antibacterial therapy. However, there are challenges associated with the use of PDT to effectively treat biofilm infections, as the ROS generated by photosensitizers (PS) can be depleted by the high levels of GSH in biofilms. To address these issues, most studies focus on the depletion of endogenous GSH by mediators such as nitric oxide (NO)." However, the strategy to exploit high GSH levels in biofilms to enhance PDT have barely been developed. In this study, we utilized the high GSH levels present in biofilms, which react with SeC@PA to generate hydroxyl radicals ($\bullet\text{OH}$). Additionally, SeC@PA has the ability to deplete GSH and produce RNS. This self-enhancing strategy of generating more reactive species (RS) and decreasing GSH doubly amplified the anti-biofilm effect of PDT, which provides a novel biofilm microenvironment triggered self-enhancing strategy for the treatment of biofilm infections.

(2) Bidirectional regulation of RS exploiting the different GSH levels between biofilms and

wound tissue is a crucial aspect that we highlight in our manuscript (Page 3): “Another point of concern is that ROS is a double-edged sword. Wound beds covered by biofilms are excessively inflammatory, and oxidative stress in wounds can be aggravated by the ROS produced during PDT used to destroy biofilms, leading to impaired wound healing. Therefore, the prompt elimination of ROS during PDT to prevent further aggravation of wound inflammation, and increasing the antioxidative capacity of the wound beds to endow upon wounds an anti-inflammatory effect from the outside to the inside are of utmost importance.” In our work, we employed SeC@PA to exploit the different GSH levels in wound tissue and biofilms enabling bidirectional regulation of RS (including ROS and RNS). In the wound tissue with low GSH levels, SeC@PA functions to clear RS and enhances the content and activity of GPX4, thereby exhibiting a strong antioxidant capacity.

To further support the significance of bidirectional RS regulation, we conducted additional experiments. Firstly, we measured GSH concentrations in wound tissue and wound coverages, including biofilms and inflammatory exudates. The results, presented in Supplementary Fig. 26, demonstrated a significant difference in GSH levels between the two sites. The GSH content in the wound coverings was significantly higher than that in the wound tissue, with a content as high as 6.69 mmol/g, which was 9.3 times higher than 0.72 mmol/g in the tissue. These results make it possible for SeC@PA to bidirectionally regulate RS between biofilms and wound tissue.

Secondly, we employed a diabetic wound model without biofilms infection to verify the direct damaging effect of PDT on wound tissue. As shown in Figure R9, the C@PA (+) group seriously hindered the wound healing, attributed to the more serious oxidative stress at the wound. On the contrary, the SeC@PA (+) group significantly promoted wound healing and significantly reduced oxidative stress in wound tissue. These findings highlight the potential side effects of RS-induced oxidative stress during PDT treatment for chronic wounds with biofilm infection. In our work, this issue was addressed by the bidirectional regulatory effect of SeC@PA on RS. This bidirectional regulation strategy provides a new way for the wound treatment of biofilm infection.

(3) Optimized delivery strategy. As we state in original manuscript (Page 3): “Moreover, an important factor affecting the efficacy of therapeutics is their point of delivery. The importance of the role of the delivery point in promoting wound healing has been well studied. Chronic wounds are typically covered with eschar and biofilms, which constitute a physiochemical barrier hindering the penetration of topical therapeutic agents into the underlying live tissue. Thus, when drugs are used topically, their local bioavailability is reduced, as expected. A delivery system that can disrupt the physiochemical barriers created by eschar and biofilms and shorten the travel distance of therapeutics by directly delivering them to live tissue can improve drug efficacy.” In our work, microneedles facilitate to break the barrier formed by biofilms and inflammatory exudates, contributing to the effective delivery of SeC@PA. We compared the therapeutic effects of SeC@PA MN and SeC@PA NPs, as well as their

penetration efficiency into wound coverings (Fig. 7), to underscore the significance of microneedles in the treatment of chronic wounds with biofilm infection.

Overall, this work highlights the self-enhancement strategy triggered by biofilm microenvironment, bidirectional RS regulation strategy, and optimized delivery strategy to address the current bottlenecks in PDT treatment of biofilm-infected chronic wounds.

Supplementary Fig. 26. GSH content of tissue and wound coverings (n=5, mean \pm SD). ****P<0.0001 compared with the indicated group.

Figure R9. Photodynamic therapy alone delays wound healing. (A, C) Representative images of wounds during healing and quantitative data of relative wound area to day 9 of the different groups at different time points (n=6, mean ± SD). Scale bar is 5 mm. *P<0.05 compared with the indicated group, #P<0.05 compared with other groups. **(B, D)** Representative immunofluorescence images of RS staining and relative fluorescence intensity of RS at day 2 (n=3, mean ± SD). Scale bar is 100 μm. *P<0.05 compared with the other groups.

Fig. 7 SeC@PA MN for promoting chronic wound healing through efficient delivery of SeC@PA NPs. a, g Representative images of wounds during healing and quantitative data of relative wound area to day 16 of the different groups at different time points ($n=4$, mean \pm SD). Scale bar is 5 mm. $*P<0.05$ compared with the indicated group. **b** Representative H&E staining images of wound samples treated with different groups on day 16. Scale bar is 100 μm . **c, h** Representative Masson's trichrome-stained images of wound samples and quantification of the collagen volume fraction for four groups on day 16 ($n=3$, mean \pm SD). Scale bar is 500 μm . $**P<0.01$ compared with the indicated group. **d, i** Representative immunofluorescence images of CD31 staining in the regenerated skin tissue and quantification of blood vessel density on day 9 after wound healing ($n=3$, mean \pm SD). Scale bar is 100 μm . $**P<0.01$ compared with the indicated group. **e** Representative images of distribution of SeR@NPs in wound coverings at different time in SeR@PA MN group. Scale bar is 500 μm . **f** Representative images of distribution of SeR@NPs in wound coverings at different time in SeR@PA NPs group. Scale bar is 500 μm . **j** Schematic representation of the acquisition of ex vivo wound coverages and determination of penetration efficiency of SeR@PA in wound coverings. **k** Penetration efficiency of SeR@PA MN and SeR@PA NPs in wound coverings ($n=3$, mean \pm SD). SeC@PA NPs(+): Solution containing Se-Ce6-PDA-LA nanoparticles under 660 nm irradiation (200 mW/cm²) for 3 min; SeC@PA MN(+): Microneedle containing Se-Ce6-PDA-LA nanoparticles

under 660 nm irradiation (200 mW/cm²) for 3 min; SeR@PA NPs: Solution containing Se-Rhodamine-PDA-LA nanoparticles; SeR@PA MN: Microneedle containing Se-Rhodamine-PDA-LA nanoparticles.

To better exhibit our novelty, we have modified the manuscript, as followed:

We added treatment of non-biofilm infected wounds in diabetic mice using SeC@PA bandages as **Fig. 6**.

We added SeC@PA MN promoted chronic wound healing by efficient delivery of SeC@PA as **Fig. 7**.

In Page 18, we added the statement “Diabetic chronic wounds without infected biofilm were employed to verify the direct RS-scavenging....”

In Page 19, we added the statement “Chronic diabetic wounds are mostly covered by inflammatory exudates and biofilms,”

Question 2: The claims that the material is suitable for less inflamed and non-infected animals are not tested. Proper animal studies should be performed to investigate the benefit of the MNs in such animal models.

Response: Thank you for your advice. To explore the benefits of the microneedles prepared in this study on less inflamed and non-infected wound models, a wound model of diabetic mice with non-biofilm infection was constructed. As shown in Fig. 6a and f, the wound healing rate of the microneedle groups containing Se (Se@PA MN, Se@PA MN(+), SeC@PA MN, SeC@PA MN(+)) was significantly higher than that of the microneedle groups without Se (Control, Control(+), C@PA MN, C@PA MN(+)) due to the powerful antioxidant capacity of Se. Predictably, wound healing was significantly impaired in the C@PA MN (+) group, which was due to the reason that RS produced aggravated oxidative stress in the wound (Fig. 6b,g). Interestingly, the level of RS in the SeC@PA MN (+) group was much lower than that in the C@PA MN (+) group, indicating that SeC@PA MN (+) played a role in scavenging RS in wound tissue. Furthermore, Hematoxylin-eosin (HE) staining (Fig. 6c), Masson's trichrome staining (Fig. 6d,h) and CD31 immunofluorescence (Fig. 6e,i) showed that the Se containing microneedle groups exhibited a more regular skin structure, significantly enhanced collagen deposition and increased blood vessels density. Collectively, in the diabetic chronic wound model without biofilm infection, Se contained microneedles reduced inflammation by clearing RS, promoted collagen deposition and angiogenesis in the wound site for facilitating wound healing.

These results indicated that RS produced by PDT treatment alone (C@PA MN(+)) has the risk of delaying diabetic chronic wound healing, as we mentioned in the original manuscript (page 3). The microneedle we prepared with bidirectional regulation of oxidative stress can effectively avoid this risk. It can promote the healing of diabetic chronic wounds with biofilm infection or non-biofilm infection. These results are expected to extend the application of

SeC@PA MN.

Fig. 6 SeC@PA MN for promoting the healing of non-biofilm infected full-thickness diabetic wounds in rats. **a, f** Representative images of wounds during healing and quantitative data of relative wound area to day 9 of the different groups at different time points ($n=6$, mean \pm SD). Scale bar is 5 mm. $*P<0.05$ compared with the indicated group, $\#P<0.05$ compared with other groups. **b, g** Representative immunofluorescence images of RS staining and relative fluorescence intensity of RS at day 2 ($n=3$, mean \pm SD). Scale bar is 100 μm . $*P<0.05$ compared with the indicated groups. **c** Representative H&E staining images of wound samples treated with different groups on day 9. Scale bar is 100 μm . **d, h** Representative Masson's trichrome-stained images of wound samples and quantification of the collagen volume fraction for four groups on day 9 ($n=3$, mean \pm SD). Scale bar is 500 μm . $*P<0.05$ compared with the indicated group. **e, i** Representative immunofluorescence images of CD31 staining in the regenerated skin tissue and quantification of blood vessel density on day 9 after wound healing ($n=3$, mean \pm SD). Scale bar is 100 μm . $*P<0.05$ compared with the indicated group. Control: Blank

microneedle; Control(+): Blank microneedle under 660 nm irradiation (200 mW/cm²) for 3 min; C@PA MN: Microneedle containing Ce6-PDA-LA nanoparticles; C@PA MN(+): Microneedle containing Ce6-PDA-LA nanoparticles under 660 nm irradiation (200 mW/cm²) for 3 min; Se@PA MN: Microneedle containing Se-PDA-LA nanoparticles; Se@PA MN(+): Microneedle containing Se-PDA-LA nanoparticles under 660 nm irradiation (200 mW/cm²) for 3 min; SeC@PA MN: Microneedle containing Se-Ce6-PDA-LA nanoparticles; SeC@PA MN(+): Microneedle containing Se-Ce6-PDA-LA nanoparticles under 660 nm irradiation (200 mW/cm²) for 3 min.

To highlight the advantages of SeC@PA MN, we have modified the manuscript as followed:

We added the experiment of SeC@PA MN promoted the healing of non-biofilm infected full-thickness diabetic wounds in rats as **Fig. 6**.

In Page 18, we added the statement “Diabetic chronic wounds without infected biofilm were employed to verify the direct RS-scavenging....”

Question 3: The *in vitro* results and the animal studies show very limited difference between different groups. It is not clear why authors believe the final design is more successful.

Response: thank you very much for your valuable comment. *In vitro* anti-biofilm assay, anti-inflammatory assay, cell migration and tube formation assay showed that SeC@PA not only had excellent anti-biofilm ability, but also could clear RS, increase the content and activity of GPX4, exert a strong anti-inflammatory effect, and promote the migration and tube formation of macrophages by polarizing macrophages into M2 type. Although C@PA(+) also exhibited good anti-biofilm ability, they lacked anti-inflammatory effects and the ability to promote cell migration and tube formation. Additionally, C@PA(+) increased RS level in wound tissue, leading to cellular damage and cell function impairment. Similarly, Se@PA was unable to resist biofilm despite its antioxidant and M2 macrophage polarization abilities. The results of *in vivo* experiments showed that only SeC@PA MN (+) had the effect of promoting wound healing.

The anatomy of the wound tissue after the treatment (Figure R9) showed that compared with other groups, SeC@PA (+) had no obvious bleeding, redness and swelling, indicating a more favorable healing process that resembled normal tissue structure. Histopathological analysis of the regenerated skin further supported these observations, revealing that the SeC@PA MN (+) group had a more regular skin structure, increased collagen deposition, and more obvious angiogenesis. Based on the combined *in vitro* and *in vivo* results, it was concluded that SeC@PA MN(+), combined with anti-biofilm, anti-inflammatory and pro-angiogenic abilities, could effectively accelerate the chronic wound healing.

Figure R11. Representative images of wound anatomy in different groups at day 16 (n=3).

Question 4: The benefit of MNs are not demonstrated. Also, the MNs are small and typically cannot penetrate biofilms in the depth of tissue in humans. Authors should describe how this potentially can be used in real wounds. Also, the benefit of MNs over topical treatment should be demonstrated. Also, the effect of photo illumination alone should be tested. A group studying the effect of materials without photo-illumination should also be studied.

Response: Thanks a lot for your professional questions.

(1) For the first question “Also, the MNs are small and typically cannot penetrate biofilms in the depth of tissue in humans. Authors should describe how this potentially can be used in real wounds.”

SeC@PA MN prepared in this work was used for the management of diabetic chronic wounds. Most diabetic chronic wounds are open wounds and infected by biofilms. According to previous study (*Adv. Mater.* 2023, 35, 2208069), the thickness of a typical SA biofilm *in vivo* is 150 μm above the wound surface and 190 μm below the wound surface. The length of the prepared SeC@PA MN tip is more than 500 μm . Therefore, penetration depth of SeC@PA MN was higher than the typical thickness of *in vivo* SA biofilm infections in skin wounds. The size of the SeC@PA MN prepared in this work was 1.2 cm \times 1.2 cm. If the patient has a small wound area (nor large than 1.2 cm \times 1.2 cm), a microneedle bandage can be directly applied for treatment. In the case of a larger wound area, multiple microneedle bandages can be used simultaneously to ensure the entire wound is completely covered with microneedles. In fact, combined with 3D technology, the size, tip length and shape of microneedles can be personalized-designed according to the needs (*Adv. Healthcare Mater.* 2022, 11, e2102659; *Biomater. Sci.* 2023, 11, 583; *J. Control. Release* 2021, 329, 907), which enhances the utility of microneedle bandage in the wound field.

(2) For the second question “Also, the benefit of MNs over topical treatment should be demonstrated.”

To demonstrate the advantages of microneedle in the treatment of diabetic chronic wounds, we compared the therapeutic effects of SeC@PA MN(+) with SeC@PA NPs(+). Specifically, the same amount of SeC@PA NPs in SeC@PA MN was applied directly onto the wounds. The wounds were then gently sealed with a medical bandage for 24 h to prevent the loss of SeC@PA NPs during the mouse’s activity. In the case of SeC@PA MN group, the MNs were applied to the wound for 10 min before the backing was removed. To maintain the consistency of the experiments, the wounds of the control group and SeC@PA MN group were similarly sealed with a medical bandage for 24 h. As shown in Fig. 7, the SeC@PA MN(+) group demonstrated a good effect in promoting wound healing (Fig. 7a,g) with higher collagen deposition fraction (7c,h) and vessel density (7d,i) compared to the control group and SeC@PA NPs(+) group. However, there was no obvious difference between the SeC@PA NPs(+) group and control group in terms of promoting wound healing.

To investigate the position of SeC@PA after administration and its impact on therapeutic effects, we investigated the distribution of SeC@PA after administration. Firstly, chronic diabetic wounds were infected with *E. coli* expressing green fluorescent protein (GFP) to create a biofilm for easier observation. After two days of biofilm growth, the intact wound coverings, which contained both the inflammatory exudate and the biofilms, was carefully removed for subsequent experiments.

Secondly, rhodamine was used instead of Ce6 to enhance the visibility of SeC@PA distribution. As shown in Fig. 7e,f, SeR@PA MN effectively punctured the wound coverings and delivered SeR@PA to the underside of the wound coverings. The penetration depth of SeR@PA was increased with the duration of administration. However, SeR@PA from SeR@PA NPs group was mainly captured by the biofilms on the surface of the wound coverings, which could neither reach the biofilms deep in the wound coverings nor the area below it. Subsequently, we placed the dissected wound coverings into transwells, which were then inserted into a 24-well plate to investigate the penetration efficiency of SeR@PA MN and SeR@PA NPs into the wound coverings (Fig. 7j).

The results of the penetration efficiency of SeR@PA MN and SeR@PA NPs into the wound coverings were shown in Fig. 7k, indicating that the SeR@PA MN group exhibited higher penetration efficiency compared to the SeR@PA NPs group. At the same time, we performed bacterial coated plate counting on the wounds after 4 days of treatment with SeC@PA MN(+) and SeC@PA NPs (+). The results shown in Supplementary Fig. 27 indicated a stronger antibacterial effect of SeC@PA MN(+) compared to that of SeC@PA NPs(+) group.

These results demonstrated the unique advantages exhibited by microneedles in the treatment of chronic wounds. Chronic diabetic wounds are often covered with inflammatory exudates and biofilms, which create a physical barrier hindering the penetration of therapeutic drugs to the action sites beneath the covering. Therefore, traditional local delivery methods

have limitations in the treatment of chronic wounds. On the contrary, microneedles can penetrate the physical barrier of the wound surface enabling deeper and wider drug delivery. This allows for effective eradication of biofilm, reduction of oxidative stress, promotion of angiogenesis and deposition of collagen in the wound. Ultimately, these mechanisms contribute to the effective promotion of wound healing.

Fig. 7 SeC@PA MN for promoting chronic wound healing through efficient delivery of SeC@PA NPs. a, g Representative images of wounds during healing and quantitative data of relative wound area to day 16 of the different groups at different time points ($n=4$, mean \pm SD). Scale bar is 5 mm. $*P<0.05$ compared with the indicated group. **b** Representative H&E staining images of wound samples treated with different groups on day 16. Scale bar is 100 μm . **c, h** Representative Masson's trichrome-stained images of wound samples and quantification of the collagen volume fraction for four groups on day 16 ($n=3$, mean \pm SD). Scale bar is 500 μm . $**P<0.01$ compared with the indicated group. **d, i** Representative immunofluorescence images of CD31 staining in the regenerated skin tissue and quantification of blood vessel density on day 9 after wound healing ($n=3$, mean \pm SD). Scale bar is 100 μm . $**P<0.01$ compared with the indicated group. **e** Representative images of distribution of SeR@NPs in wound coverings at different time in SeR@PA MN group. Scale bar is 500 μm . **f** Representative images of distribution of SeR@NPs in wound coverings at different time in SeR@PA NPs group. Scale bar

is 500 μm . **j** Schematic representation of the acquisition of ex vivo wound coverages and determination of penetration efficiency of SeR@PA in wound coverings. **k** Penetration efficiency of SeR@PA MN and SeR@PA NPs in wound coverings ($n=3$, mean \pm SD). SeC@PA NPs(+): Solution containing Se-Ce6-PDA-LA nanoparticles under 660 nm irradiation (200 mW/cm^2) for 3 min; SeC@PA MN(+): Microneedle containing Se-Ce6-PDA-LA nanoparticles under 660 nm irradiation (200 mW/cm^2) for 3 min; SeR@PA NPs: Solution containing Se-Rhodamine-PDA-LA nanoparticles; SeR@PA MN: Microneedle containing Se-Rhodamine-PDA-LA nanoparticles.

Supplementary Fig. 27. Bactericidal results at day 4 characterized by the standard plate counting assay ($n=3$, mean \pm SD). ** $P<0.01$ and * $P<0.05$ compared with the indicated group.

(3) For the third question “Also, the effect of photo illumination alone should be tested. A group studying the effect of materials without photo-illumination should also be studied.”

We have added the study of the effect of photo illumination alone (Control(+) group and Se@PA MN(+)) and materials without photo-illumination (C@PA MN group, SeC@PA MN group) in Supplementary Fig. 28. Consistent with the results obtained in the original manuscript (Fig. 8), only SeC@PA MN (+) displayed accelerated wound healing. There was no obvious difference between the control group and the other groups in accelerating wound healing. Although the Se containing groups (Se@PA MN (+) and SeC@PA MN) effectively increased collagen deposition and angiogenesis, this degree of increase did not show any obvious effect on accelerating wound healing, as we found in our original manuscript (Fig. 8d, e, g and h). Different from non-biofilm infected diabetic chronic wounds, effective anti-inflammation can significantly accelerate wound healing. However, chronic diabetic wounds infected with biofilms is more complex. PDT alone can aggravate the oxidative stress in the

wound, while single anti-inflammatory strategy is insufficient to eradicate the biofilm. Therefore, for diabetic chronic wounds with biofilm infection, an integrated approach combining anti-inflammatory and anti-biofilm strategy could achieve satisfactory therapeutic effects. To demonstrate our point more intuitively, we selected the following groups for the subsequent experiment: Control group, C@PA MN(+) (representing PDT anti-biofilm monotherapy), Se@PA MN (representing anti-inflammatory monotherapy), and SeC@PA MN(+) (representing both anti-biofilm and anti-inflammatory dual treatment).

Supplementary Fig. 28 SeC@PA MN for promoting the healing of biofilm-infected full-thickness diabetic wounds in rats. **a, e** Representative images of wounds during healing and quantitative data of relative wound area to day 16 of the different groups at different time points ($n=6$, mean \pm SD). Scale bar is 5 mm. $*P<0.05$ compared with the indicated group. **b** Representative H&E staining images of wound samples treat with different groups on day 16. Scale bar is 100 μm . **c, f** Representative Masson's trichrome-stained images of wound samples and quantification of the collagen volume fraction for four groups on day 16 ($n=3$, mean \pm SD). Scale bar is 500 μm . $**P<0.01$ compared with the indicated group. **d, g** Representative immunofluorescence images of CD31 staining in the regenerated skin tissue and quantification of blood vessel density on day 16 after wound healing ($n=3$, mean \pm SD). Scale bar is 100 μm . $**P<0.01$ compared with the indicated group. Control: Blank microneedle; Control(+): Blank microneedle under 660 nm irradiation ($200 \text{ mW}/\text{cm}^2$) for 3 min; C@PA MN: Microneedle containing Ce6-PDA-LA nanoparticles; C@PA MN(+): Microneedle containing Ce6-PDA-LA nanoparticles under 660 nm irradiation ($200 \text{ mW}/\text{cm}^2$) for 3 min; Se@PA MN: Microneedle containing Se-PDA-LA nanoparticles; Se@PA MN(+): Microneedle containing Se-PDA-LA nanoparticles under 660 nm irradiation ($200 \text{ mW}/\text{cm}^2$) for 3 min; SeC@PA MN: Microneedle containing Se-Ce6-PDA-LA nanoparticles; SeC@PA MN(+): Microneedle containing Se-Ce6-PDA-LA nanoparticles under 660 nm irradiation ($200 \text{ mW}/\text{cm}^2$) for 3 min.

Fig. 8 SeC@PA MN for promoting the healing of biofilm-infected full-thickness diabetic wounds in rats. a Representative images of wounds during healing. Scale bar is 5 mm. **b** Schematic diagram of the wound healing process of the four groups. **c** Representative H&E

staining images of wound samples treated with different groups on day 16. **d, g** Representative Masson's trichrome-stained images of wound samples and quantification of the collagen volume fraction for four groups on day 16 ($n=3$, mean \pm SD). Scale bar is 500 μm . $***P<0.001$ and $**P<0.01$ compared with the indicated group. **e, h** Representative immunofluorescence images of CD31 staining in the regenerated skin tissue and quantification of blood vessel density on day 16 after wound healing ($n=3$, mean \pm SD). Scale bar is 100 μm . $**P<0.01$ and $*P<0.05$ compared with the indicated group. **f** Quantitative data of relative wound area to day 16 of the four groups at different time points ($n=6$, mean \pm SD). $***P<0.001$, $**P<0.01$ and $*P<0.05$ compared with the indicated group. Control: Blank microneedle; C@PA MN(+): Microneedle containing Ce6-PDA-LA nanoparticles under 660 nm irradiation (200 mW/cm²) for 3 min; Se@PA MN: Microneedle containing Se-PDA-LA nanoparticles; SeC@PA MN(+): Microneedle containing Se-Ce6-PDA-LA nanoparticles under 660 nm irradiation (200 mW/cm²) for 3 min.

Question 5: Figure captions should be expanded and be stand alone. The groups should be described in the captions.

Reponses: Thank you for your suggestion. We have added the description of the experimental groups in the figure captions. In order to provide a clearer description of the experimental groups, we have added the relevant information to the figure captions.

Question 6: Most chronic wounds are basic. Authors should also perform the in vitro studies in a basic environment.

Response: Thank you for your advice. Although chronic wounds typically have a basic microenvironment, the action of SeC@PA occurs within the biofilms and cells, which both are acidic microenvironment. Specifically, SeC@PA penetrates the biofilms and releases Se (Supplementary Fig. 5) and Ce6 (Supplementary Fig. 4) in the acidic microenvironment, generating "RS storm" that effectively eradicates the biofilms. The acidic nature of the biofilm microenvironment has been documented in previous literature (e.g., *Adv. Mater.* 2023, 19, e2211330; *Adv. Funct. Mater.* 2022, 32, 2209185; *ACS Nano* 2022, 14, 347). Once taken up by living cells in the wound bed, SeC@PA, prepared based on polydopamine, is transported to the lysosome with a pH range of 4-5 (*J. Control. Release* 2020, 317, 232; *Nat. Nanotechnol.* 2020, 15, 252). In the lysosome, the polydopamine layer degrades, releasing Se, which exerts antioxidant and immunomodulatory functions. The dopamine-based spherical nanoparticles are internalized by cells and subsequently enter the lysosome, and the degradation of the polydopamine layer in lysosomes has been reported (*J. Control. Release* 2020, 317, 232; *Nano Lett.* 2017, 11, 6790). Importantly, our experimental results validated these perspectives. Following the penetration of SeC@PA into the biofilms, the intensity of RS within the biofilms significantly increased (Fig. 3b and Supplementary Fig. 15b). However, upon uptake by cells (including HUVECs and RAW264.7 cells), SeC@PA led to a notable reduction in cellular RS

levels (Fig. 4a-c). Additionally, the activity and content of GPX were obviously increased (Fig. 4d-g), and macrophages undergo polarization towards the M2 type (Fig. 5a). Notably, if a basic microenvironment (pH 7-9) is employed as the pH condition *in vitro*, the release rate of Se and Ce6 from SeC@PA would be much lower compared to weakly acidic conditions (Supplementary Fig. 4 and Table R1). This discrepancy fails to accurately simulate the actual release behavior of SeC@PA at the site of action (biofilms and live cells) *in vivo*. Therefore, we selected an acidic microenvironment with a pH of 5.5 to represent the acidic conditions for our *in vitro* experiments.

Supplementary Fig. 4 Ce6 release profile in PBS with different pH values ($n=3$, mean \pm SD).

Supplementary Fig. 5 Se release profile in PBS at pH = 5.5 ($n=3$, mean \pm SD).

Fig. 3b Detection of RS in the SA biofilm incubated with different nanoparticles after 660 nm irradiation (200 mW/cm²) for 3 min. Scale bar is 500 μm.

Supplementary Fig. 15b Detection of RS in the PA biofilm incubated with different nanoparticles after 660 nm irradiation (200 mW/cm²) for 3 min. Scale bar is 500 μm

Fig. 4 Anti-inflammatory effect of nanoparticles. **a, b** Detection of RS by fluorescence of DCF in the HUVECs incubated with different nanoparticles under 660 nm irradiation (200 mW/cm²) for 3 min. Scale bar is 500 μm. ****P*<0.001 compared with the indicated group. **c** Intracellular DCF fluorescence intensity of HUVECs with different groups analyzed by flow cytometry. **d** Representative images of GPX4 staining of HUVECs after different treatments. Scale bar is 200 μm. **e, f** Expression level of GPX4 in the HUVECs determined by western blotting after treatments (*n* = 3, mean ± SD). **P*<0.05 and ***P*<0.01 compared with the indicated group. **g** Activity of GPX4 treated with different groups (*n* = 3, mean ± SD). ****P*<0.01 compared with the indicated group. Se@PA: Se-PDA-LA nanoparticles; C@PA(+): Ce6-PDA-LA nanoparticles under 660 nm irradiation (200 mW/cm²) for 3 min; SeC@PA(+): Se-Ce6-PDA-LA nanoparticles under 660 nm irradiation (200 mW/cm²) for 3 min.

Fig. 5a Flow cytometry analysis of F4/80⁺ CCR7⁺ and F4/80⁺ CD206⁺ cells.

Table R1 Cumulative release of Se from SeC@PA at 48 h under different pH conditions.

pH=5.5	pH=7.4
56.39 %	7.67 %

reviewers' Comments:

Reviewer #1:

Remarks to the Author:

The authors have carefully revised the manuscript and addressed all the issues. I agree to the acceptance of this manuscript.

Reviewer #2:

Remarks to the Author:

Most revised answers were up to the accept level, but there are still some minimum issues that need to be addressed before this article can be published.

1. In the reply to Question 2, the author's text describes the unit of GSH as mM but the unit in Figure 2e is μM . The author double check and keep consistence.
2. "Question 7: The anti-biofilm property of PA is lack of in-depth discussion. As shown in Figure 3 and S10, the anti-biofilm property of PA is weaker than that of SA, such phenomenon should be explained." Authors failed to explain the reason, which could cause readers confused. This explanation should be added in discussion.
3. "When the GSH concentration is low, the production of RS is limited and could be scavenged by Se and GSH in the system." Please explain why Se can clear RS.
4. "However, the strategy of exploiting high GSH level in biofilms to enhance PDT have been barely reported. In this study, we proposed a strategy to effectively amplify PDT by depleting endogenous GSH, and generate active molecules that are more lethal to biofilms by utilizing endogenous GSH, thus facilitating the eradication of biofilms." This part should give reference. DOI: 10.1002/sml.202302547. is good example for this strategy.
5. There is a lot of blank space between the pictures in the article, please use the space in the picture efficiently.

Reviewer #3:

Remarks to the Author:

The authors have addressed most of the concerns. While the level of innovation is moderate, they have demonstrated the benefit of the microneedle patch in vivo. The response to the comparison between topical delivery and MN-mediate delivery is not satisfactory. Overall, the manuscript can be published in Nature Comm.

Point-by-Point Response to Reviewers

To Reviewer #1 (Remarks to the Author):

The authors have carefully revised the manuscript and addressed all the issues. I agree to the acceptance of this manuscript.

Response: We thank the reviewer for the recommendation of publication.

To Reviewer #2 (Remarks to the Author):

Most revised answers were up to the accept level, but there are still some minimum issues that need to be addressed before this article can be published.

Response: We thank the reviewer for the further evaluation of our manuscript, and we have revised the manuscript according to the advice as shown below.

1. In the reply to Question 2, the author's text describes the unit of GSH as mM but the unit in Figure 2e is μM . The author double check and keep consistence.

Response: We appreciate the reviewer's attention to detail in helping us identify this error. We have made the necessary changes to the image, correcting " μM " to "mM".

2. "Question 7: The anti-biofilm property of PA is lack of in-depth discussion. As shown in Figure 3 and S10, the anti-biofilm property of PA is weaker than that of SA, such phenomenon should be explained." Authors failed to explain the reason, which could cause readers confused. This explanation should be added in discussion.

Response: We apologize for the previous incomplete response to Question 7. We speculate that the reviewer's assessment of SeC@PA(+) having a weaker effect on the PA biofilms compared to the SA biofilms was based on the following two points: 1. The intensity of RS generated by the SeC@PA(+) group in the SA biofilms in Fig. 3b appears to be stronger than that in the PA biofilms in Supplementary Fig. 15b; 2. There are no bacterial colonies on the plate in the SeC@PA(+) group in Fig. 3c, while bacterial colonies are still present in Supplementary Fig. 15c.

In fact, when investigating the eradication effect of SeC@PA(+) on the PA biofilms and SA biofilms, we conducted two independent experiments. Therefore, we can only ensure that the experimental parameters were consistent and comparable among all groups when exploring the SA biofilms or PA biofilms. However, the investigation of the SA biofilms was not conducted simultaneously with the investigation of the PA biofilms, and these two independent experiments cannot be directly compared due to the differences in experimental parameters.

Below, we present the raw data for live/dead biomass ratios and plate counting results of the control group and SeC@PA(+) groups for both SA biofilms and PA biofilms. The data have been normalized relative to their respective control groups for comparison. The live/dead biomass ratio is normalized as a percentage, while the plate counting result is normalized as a decrease in magnitude. The data analysis showed that after data normalization, both the live/dead biomass ratio and plate counting results indicated no significant difference in the eradication effects of the SeC@PA(+) group on the SA biofilms and PA biofilms. Therefore, based on these experimental results, we suggest that SeC@PA(+) has comparable anti-biofilm effects on PA biofilms and SA biofilms.

Table R1. Live/dead biomass ratio of control group and SeC@PA(+) group in different biofilms.

Group	SA biofilm				PA biofilm	
Control	99.7	99.5	94.5	100.0	99.6	99.5
SeC@PA(+)	3.2	4.8	0.3	4.0	12.7	1.6
Normalization	3.3	4.9	0.3	4.0	12.7	1.6

Figure R1. Live/dead biomass ratio normalized results of different biofilms.

Table R2. Plate counting result of control group and SeC@PA(+) group in different biofilms.

Group	SA biofilm				PA biofilm	
Control	55000000	73000000	89000000	197000000	236000000	207000000
SeC@PA(+)	900	1900	1200	30800	29100	26300
Normalization	4	4	3	4	4	4

Figure R2. Plate counting normalized results of different biofilms.

3. “When the GSH concentration is low, the production of RS is limited and could be scavenged by Se and GSH in the system.” Please explain why Se can clear RS.

Response: Thanks for the comments. "Se" refers to nanoscale elemental selenium obtained by the reduction of selenite with vitamin C. Due to its small particle size and large surface area, it exhibits good RS scavenging capabilities. In fact, several studies have demonstrated the ability of Se to remove RS (e.g., *ACS Nano* 2022, 16, 18667; *Adv. Healthcare Mater.* 2023, 12, e2203160). In this study, we confirmed through X-ray photoelectron spectroscopy (XPS) that Se can be oxidized by RS to a higher valence state, thereby reducing the RS and achieving the function of RS scavenging (Fig. 2f).

Fig. 2f High-resolution Se 3d XPS spectra of SeC@PA(+).

4. “However, the strategy of exploiting high GSH level in biofilms to enhance PDT have been barely reported. In this study, we proposed a strategy to effectively amplify PDT by depleting endogenous GSH, and generate active molecules that are more lethal to biofilms by utilizing endogenous GSH, thus facilitating the eradication of biofilms.” This part should give reference. DOI: 10.1002/smll.202302547. is good example for this strategy.

Response: We would like to express our gratitude to the reviewer for providing the valuable reference. After carefully reviewing it, we found that it is an excellent example of utilizing the high GSH concentration in biofilms to amplify PDT, which is relevant to the strategies discussed in our manuscript. We have appropriately cited this reference in the manuscript.

5. There is a lot of blank space between the pictures in the article, please use the space in the picture efficiently.

Response: Thanks for the reviewer's valuable suggestion. We have made the necessary adjustments to the image layout, effectively utilizing the available blank space.

To Reviewer #3 (Remarks to the Author):

The authors have addressed most of the concerns. While the level of innovation is moderate, they have demonstrated the benefit of the microneedle patch in vivo. The response to the comparison between topical delivery and MN-mediate delivery is not satisfactory. Overall, the manuscript can be published in Nature Comm.

Response: We are extremely grateful for the evaluation and the assistance provided by the reviewer in improving the quality of our manuscript. We also thank the reviewer for the recommendation of publication.

Reviewers' Comments:

Reviewer #2:

Remarks to the Author:

After this round of revisions, the article is allowed to accepted.

Response to Reviewers' Comments

Reviewer #2 (Remarks to the Author):

After this round of revisions, the article is allowed to accepted.

Our response: We appreciate the reviewer's recommendation of publication.